# Advances in the Optimization of Fe Nanoparticles: Unlocking Antifungal Properties for Biomedical Applications

**DOI:** 10.3390/pharmaceutics16050645

**Published:** 2024-05-10

**Authors:** Zeshan Ali Sandhu, Muhammad Asam Raza, Abdulmajeed Alqurashi, Samavia Sajid, Sufyan Ashraf, Kainat Imtiaz, Farhana Aman, Abdulrahman H. Alessa, Monis Bilal Shamsi, Muhammad Latif

**Affiliations:** 1Department of Chemistry, Faculty of Science, Hafiz Hayat Campus, University of Gujrat, Gujrat 50700, Pakistan; zeshansandhu89@gmail.com (Z.A.S.); sufyanarmin9@gmail.com (S.A.); kainatimtiaz242@gmail.com (K.I.); 2Department of Biology, College of Science, Taibah University, Madinah 42353, Saudi Arabia; aqurashi@taibahu.edu.sa; 3Department of Chemistry, Faculty of Science, University of Engineering and Technology, Lahore 54890, Pakistan; mayachaudhary022@gmail.com; 4Department of Chemistry, The University of Lahore, Sargodha Campus, Sargodha 40100, Pakistan; farhanaamman@gmail.com; 5Department of Biology, Faculty of Science, University of Tabuk, Tabuk 71491, Saudi Arabia; aalessa@ut.edu.sa; 6Centre for Genetics and Inherited Diseases (CGID), Taibah University, Madinah 42353, Saudi Arabia; monisbilalshamsi@gmail.com; 7Department Basic Medical Sciences, College of Medicine, Taibah University, Madinah 42353, Saudi Arabia

**Keywords:** nanomaterial optimization, biomedical applications, antifungal properties, biocompatibility, biomedical engineering, iron nanoparticles

## Abstract

In recent years, nanotechnology has achieved a remarkable status in shaping the future of biological applications, especially in combating fungal diseases. Owing to excellence in nanotechnology, iron nanoparticles (Fe NPs) have gained enormous attention in recent years. In this review, we have provided a comprehensive overview of Fe NPs covering key synthesis approaches and underlying working principles, the factors that influence their properties, essential characterization techniques, and the optimization of their antifungal potential. In addition, the diverse kinds of Fe NP delivery platforms that command highly effective release, with fewer toxic effects on patients, are of great significance in the medical field. The issues of biocompatibility, toxicity profiles, and applications of optimized Fe NPs in the field of biomedicine have also been described because these are the most significant factors determining their inclusion in clinical use. Besides this, the difficulties and regulations that exist in the transition from laboratory to experimental clinical studies (toxicity, specific standards, and safety concerns) of Fe NPs-based antifungal agents have been also summarized.

## 1. Introduction

In recent years, the deployment of nanoparticles (NPs) has opened new pathways for biomedical applications, resulting in novel techniques for dealing with numerous infections and diseases [1]. Iron nanoparticles (Fe NPs) have emerged as prospective contenders and have demonstrated outstanding commitment, particularly in terms of their antifungal capabilities. Fungal infections are a major health issue worldwide, particularly in immune-compromised individuals, and the rise of drug-resistant fungal species has further worsened this situation [2]. Therefore, there is an urgent need for new and more effective antifungal medicines, and Fe NPs have emerged as a possible treatment option [3]. Fe NPs are a type of nanomaterials that are characterized by their size, shape, and unique physicochemical characteristics. Owing to their biocompatibility, convenient synthesis, and diverse functionalization, these nano-scale structures have been widely investigated and used in a variety of disciplines [4]. In the field of biomedicine, Fe NPs have attracted considerable interest for many applications, ranging from targeted drug administration [5,6] to imaging [7], and most importantly, for their antimicrobial characteristics, including their potential as antifungal agents [8]. Figure 1 illustrates the key properties of the Fe NPs.

Mycoses, or fungal diseases, are major hazards to human health and agriculture. *Candida* species, such as *Aspergillus fumigatus*, and *Cryptococcus neoformans*, among others, are well-known pathogens that cause a wide range of fungal infections, ranging from superficial skin infections to life-threatening systemic illnesses [9,10]. The scarcity of effective antifungal medications, along with the increase in drug-resistant strains, has highlighted the need to develop innovative methods for fungal infection management [11,12]. Recently, Fe NPs have attracted much attention from scientists because of their superparamagnetic capabilities, good reactivity towards magnetic fields, and other distinctive characteristics, including stability and biocompatibility [13]. The utilization of Fe NPs in the biomedical field encompasses a wide range of applications, notably in the treatment of various ailments and the development of drug-delivery systems [14]. The current increase in the prevalence of invasive and potentially fatal fungal infections has raised significant concerns regarding the comparatively lower rate of identification and development of antifungal medications other than antimicrobial agents [15]. In the last 20 years, there has been an increase in fungal illnesses, especially in people with weak immune systems or in hospital [16]. The antifungal activities of Fe NPs are principally attributed to their capacity to damage fungal cell membranes, interfere with cellular processes, and generate oxidative stress, eventually leading to fungal cell death [17]. The superparamagnetic biomedical applications of Fe NPs are illustrated in Figure 2.

Fe NPs represent a viable approach for combating drug-resistant fungal populations that have evolved resistance to traditional antifungal drugs, due to these processes. Furthermore, Fe NPs may be functionalized with particular ligands or compounds to improve antifungal activity and target specific fungal species while minimizing off-target effects on healthy cells [18]. Among the vast array of fungal species, exceeding two million in number, a mere six hundred varieties have been identified as human fungal pathogens. Remarkably, only 3–4% of these species are responsible for over 99% of cases involving invasive fungal infections (IFIs) [15]. The utilization of fungicides has emerged as the predominant approach for the prevention and management of mycotoxin production. Owing to the structure-dependence of pathogenic fungi, numerous coumarin derivatives have been considered promising antifungal medicines [19].

Researchers still face many challenges associated with the current antifungal treatments. It is difficult to make an early and correct diagnosis and then use the right antifungal therapy, which causes a high death rate among people with IFI [20]. Extensive research has been conducted on the use of NPs in antifungal therapy. Fe NPs must be synthesized because they are important in the biomedical field [14]. Some studies have demonstrated notable enhancements in various drug-related characteristics, including solubility and stability in aqueous environments, enhanced bioavailability, and improved tissue penetration. These improvements have led to increased therapeutic effectiveness and diminished toxicity [21]. Fe NPs have been demonstrated to possess the potential for developing multifunctional nano-systems that could be utilized for treating fungal infections as well as serving as diagnostic and sensitizing tools, and a large number of studies have supported this assertion. The objective of this comprehensive review is to thoroughly evaluate the current advancements in the optimization of Fe NPs, with a particular focus on elucidating their potential as effective antifungal agents and their applicability in diverse biomedical contexts.

## 2. Various Synthesis Methods for Fe NPs

NPs can be prepared by chemical, physical, or biological techniques. Examples of physical techniques include gas deposition and electron beam lithography. Chemical techniques include hydrolysis [22], co-precipitation [23], sol–gel processes [24], pyrolysis [25], and chemical vapor deposition [26]. Biological methods refer to the utilization of various microbes and plant extracts to synthesize NPs [27,28]. Figure 3 shows a synthetic approach to Fe NPs.

### 2.1. Chemical Methods

The capping and reducing agents used in chemical synthesis are expensive and potentially dangerous. The medicinal and environmental use of dangerous and poisonous chemicals has negative consequences [29].

#### 2.1.1. Co-Precipitation Methodology

Among all the chemical methods, the co-precipitation method is widely employed for the synthesis of iron oxide nanoparticles (FeO NPs). The synthesis of FeO NPs involves a combination of Fe^+2^ (ferrous) and Fe^+3^ (ferric) ions in a 1:2 ratio in an alkaline solution under ambient conditions. The main benefit of this approach is its ability to produce a large number of particles with various size distributions. However, it has some drawbacks, such as difficulty in obtaining homogenous and monodisperse NPs and wastewater with a high pH, which may be harmful to the environment [30]. Al-Alawy et al. (2018) successfully manufactured Fe NPs using a co-precipitation method involving a combination of Fe^+2^ and Fe^+3^ ions in an alkaline solution. Several parameters were tested in the two-step synthesis processes, including temperature (25 and 80 °C), incorporation of a base in the reacting substances, and employing nitrogen as an inert gas [31]. Al-Madhagi et al. (2023) demonstrated that the co-precipitation method for synthesizing magnetic nanoparticles (MNPs) typically involves the utilization of two key components: iron salts, which serve as precursor materials, and a precipitating agent that aids in the precipitation of Fe_3_O_4_ NPs. These approaches were employed to attain mono-dispersity and superparamagnetism, while ensuring a diameter below 30 nm [32]. Malik et al. (2023) investigated *Nigella sativa* seeds and synthesized Fe NPs via co-precipitation. The antifungal susceptibility of the substance was subsequently assessed against *Candida albicans*, which resulted in a minimum inhibitory concentration (MIC) value of 3.125 µg/mL. Quantitative analysis was conducted to assess the levels of antioxidant defense enzymes [33]. Figure 4 depicts the facile pathway of the co-precipitation method.

#### 2.1.2. Sol–Gel Process

The sol–gel method is a widely recognized synthetic technique used for the production of Fe NPs of superior quality. This method provides exceptional control over both the texture and surface characteristics of the manufactured NPs [34]. The initial substance used in the experiment was ferric nitrate, which was mixed with ethylene glycol under continuous agitation, leading to the formation of a brown gel. The gel was allowed to stand for a duration of 2 h and subsequently subjected to a heating process at a temperature of 80 °C for approximately 4 h. Following the completion of the drying process, the gel was annealed in a temperature range of 200–400 °C in a vacuum environment [30]. Batool et al. (2023) employed the sol–gel technique to synthesize FeO NPs at pH levels ranging from 1 to 9. The process of magnetic field annealing (MFA) was conducted on NPs at two distinct temperatures, specifically 200 and 300 °C. The MFA samples exhibited efficient outcomes at a temperature of 300 °C. X-ray diffraction (XRD) analysis revealed the presence of a magnetite phase at pH values of 1, 2, and 6. Additionally, mixed phases consisting of hematite and magnetite were observed within the pH range of 3–5. At pH levels of 7–8, the XRD results indicated the presence of a magnetite phase, whereas at pH 9 the analysis revealed the presence of a hematite phase in the iron oxide samples [35]. Tadic et al. (2019) utilized the sol–gel method to produce Fe NPs in a silica matrix with a hematite content of 60%. The XRD patterns and FTIR spectra of the sample indicated the production of α in the Fe_2_O_3_ phase and amorphous silica. TEM studies revealed the presence of two particle size distributions with average diameters of 10 nm and 20 nm [36]. Marjeghal et al. (2023) synthesized ultra-fine strontium hexaferrite particles using a unique sol–gel combustion technique. By controlling the weight ratio of citric acid to strontium and iron salts nitrate (CA/MN), researchers discovered an increase in the purity of sol–gel-produced SrFe_12_O_19_ NPs [37]. The synthetic flow chart of Fe NPs via the sol–gel process is shown in Figure 5.

#### 2.1.3. Chemical Vapor Deposition (CVD)

Chemical vapor deposition is a method employed to deposit thin films via chemical reactions between vapor species and the substrate surface. The CVD technique involves the passage of a gas and precursor material through a reaction chamber maintained at a high temperature, exceeding 900 °C under vacuum conditions. The reaction transpires within a thermally controlled chamber, leading to NP synthesis [38,39]. One of the major drawbacks of this method is that, despite the high purity of the obtained material, the yield is limited, it is difficult to control various parameters, and, on the top of that, the instrumentation is expensive [40]. Atchudan et al. (2019) synthesized FeO NP-compacted multi-walled carbon nanotubes (FeO NP-MWCNT hybrid) through the direct application of the CVD process, utilizing acetylene as the carbon source. The average diameter of the manufactured FeO NPs was approximately 9 nm, whereas that of the MWCNTs was approximately 50 nm [41]. Figure 6 shows the traditional CVD method for the preparation of Fe NPs.

Ağaoğulları et al. (2017) employed a hybrid synthetic process using high-energy ball milling but low pressure, and LPCVD for the fabrication of Fe NPs, which were graphite encapsulated. LP-CVD was carried out inside a tube furnace at a pressure of about 0.57 Torr. The procedure was conducted in a controlled atmosphere of CH_4_/H_2_ at a temperature of 1050 °C for 1 to 3 h. The samples synthesized in this study were additionally treated with 2 M hydrofluoric acid (HF) solution for purification purposes [42]. Saraswati and her team (2019) developed magnetic composites comprising Fe compounds and carbon nanotubes (CNT) via CVD at different applied pressures using an FeO/C catalyst. Experimental conditions consisted of introducing a sample system to an 800 °C temperature for a period of 10 min and using nitrogen gas (N_2_) at a pressure of 20 Torr, followed by the injection of EtOH vapors, which resulted in final pressures of 80 and 100 Torr in the absence of air. However, in the presence of air, the pressure reached 180 Torr [43].

### 2.2. Physical Methods

#### 2.2.1. Laser Ablation

The laser ablation approach involves the removal of bulk material containing an iron precursor to produce FeO NPs. This process is carried out by placing the bulk material in a container containing a solvent medium that is selected and directing a laser beam with predetermined parameters, including intensity, wavelength, and dimensions, onto the bulk substance [44]. Despite being widely promoted, this approach has inherent limitations associated with the ablation procedure and technical issues. The presence of species with elevated kinetic energy levels can result in the re-sputtering phenomenon. Additionally, uneven energy distribution within the laser beam can contribute to an irregular energy profile within the plume [45]. The synthetic mechanism of the Fe NPs via laser ablation is shown in Figure 7.

Waag et al. (2022) provided experimental insights into the mixing mechanisms involved in the laser-induced production of alloy NPs. The Co-Fe alloy was subjected to experimental investigation in the presence of EtOH because it holds significance for potential utilization in oxidation catalysis [46]. Kupracz et al. (2020) successfully created Fe NPs by pulsed laser ablation. This method involves the use of a pure iron target and water [47]. Curcio et al. (2019) created iron-doped hydroxyapatite thin films. The composite target for deposition was made by combining standard hydroxyapatite powder with 4% Fe NPs generated by femtosecond laser ablation in liquid (LAL) [48].

#### 2.2.2. Mechanical Milling

Ball milling is extensively employed to alter the dimensions and structure of raw materials as well as to enhance their chemical reactivity. The time and speed at which the ball mill rotates have a significant impact on the size and characteristics of the FeO NPs produced [49]. NPs are produced via a two-step milling process. The initial stage involved the implementation of traditional ball milling. In this context, the iron particles undergo deformation and subsequently flatten, resulting in the formation of exceedingly thin flakes [50]. During the second step, the incorporation of alumina into the milling process results in fragmentation of the flakes, leading to the production of smaller INPs [51]. Zhang et al. (2020) synthesized carbon-encapsulated Fe NPs using high-energy ball milling with dopamine and ferric nitrate as the precursor materials. The significance of the elevated temperature during the procedure is believed to be pivotal in facilitating the carbonization of dopamine. Furthermore, the regional ambient temperature facilitates the reduction of Fe^3+^ ions by dopamine and the subsequent generation of NPs [49]. Darvina et al. (2019) manufactured FeO NPs using a ball-milling process with various milling times. The purity of the magnetite (Fe_3_O_4_) increased with a longer milling duration, and a homogeneous magnetite phase was achieved after 30 h of milling. The duration of milling affects various characteristics of Fe NPs, including the dimensions of crystallites, their shape, and their magnetic characteristics [52]. Seyedi et al. (2015) synthesized FeO NPs using solid-state displacement during mechanical milling. The experimental findings demonstrated that increasing the milling duration significantly decreased the size of FeO NPs. The mean diameter of the NPs was approximately 14 nm after a 2 h milling procedure, and it was reduced to 4 nm after a 5 h milling process [53]. The mechanical ball-milling process for the synthesis of the Fe NPs is illustrated in Figure 8.

#### 2.2.3. Sonication Approach

Sonochemical techniques utilize ultrasound as a fast means to achieve various objectives, such as precise manipulation of nanostructures and dispersion of materials [54,55]. Cavitation induced by ultrasonic waves has the potential to yield highly homogeneous nano-scale structures [56]. PVD involves the deposition of a substance onto a surface, resulting in the formation of thin films or NPs. The utilization of precise vacuum approaches, including thermal evaporation and sputtered deposition, induces vaporization of the material, subsequently leading to its condensation onto a substrate [56,57]. Braim et al. (2023) utilized ultrasound to efficiently modify Fe_3_O_4_ NPs by integrating their surfaces through the application of a coating consisting of bismuth (Bi NPs). This results in the formation of distinct composite NPs known as Fe_3_O_4_@Bi NPs [58]. Putri et al. (2020) used iron sand obtained from a local source to synthesize FeO NPs. This study investigated the impact of varying the sonication time and iron-sand size with the aim of achieving NPs with superior qualities. The black hue of the acquired magnetic NPs was further examined using the Brunauer–Emmett–Teller (BET) technique to investigate the porous characteristics of the material, including surface area, average pore diameter, and pore size classification [59]. Deshmukh et al. (2019) reported on an environmentally conscious method for manufacturing FeO NPs using ultrasound. Investigation of the magnetization in the FeO NPs confirmed their superparamagnetic properties. NPs manufactured using ultrasound assistance exhibited superior stability as well as enhanced antibacterial and antioxidant activities compared to NPs produced by magnetic stirring [60]. Figure 9 depicts the sonication methodology commonly used for the synthesis of Fe NPs.

### 2.3. Biological Methods (Green Synthetic Method)

NPs can be bio-reduced, formed, and stabilized using plant extracts, fungi, algae, and bacteria. The biological synthesis of NPs offers several benefits, including a reduced environmental impact, shorter production time, and lower overall costs [61].

#### 2.3.1. Plant-Mediated Synthesis of NPs

Plants possess a diverse array of biologically active substances such as alkaloids, flavonoids, terpenoids, and steroids, which serve as reducing agents during the production of NPs [61]. Plant extracts offer more advantages than microorganisms in the synthesis of NPs because of their one-step process, non-pathogenic nature, and cost-effectiveness. They facilitate the removal of potentially dangerous by-products, in addition to aiding in the precise adjustment of NP dimensions [62]. Extracts distilled from *Camellia sinensis* were used in the synthesis of FeO NPs, which exhibited not only spherical, but also cluster, formation of irregular shape [63]. In another study, the green synthesis of the Fe NPs using the extracts from a plant species was proposed by Ebrahiminezhad, and co-authors in 2018 referred to it as an eco-friendly sustainable method. It is a water-based method that builds on plant-based compounds in relation to colloids as opposed to solvents and chemicals [64]. Karimi et al. (2019) used combined extracts of plants in a green synthesis of Fe NPs; the extract was derived from *Prangos ferulacea* and *Teucrium polium* plants, and reported excellent results. Apart from the synthesis, this study found the efficiency of the obtained NPs’ capability for As (III) sorption from water or an aqueous environment [65]. It has been reported by Yusefi et al. (2020) that *Punica granatum* fruit peel extract has been acting as a stabilizer of FeO NPs production in a greener way. Yusefi et al. (2020) utilized four varying weight percentages of *Punica granatum* fruit peel extract as environmentally friendly stabilizers for the synthesis of FeO NPs. The authors subsequently performed a series of characterization and in vitro investigations, which were conducted to assess the anticancer potential of the fabricated Fe NPs. The results demonstrated that the FeO NPs exhibited a notable degree of crystallinity and purity [66]. Nadeem et al. (2021) demonstrated the synthesis of Fe NPs using *Clematis orientalis* plant extract and showed that salt concentrations modulate the physiological and biological properties of Fe NPs [67]. Plant-mediated synthesis of Fe NPs is shown in Figure 10.

#### 2.3.2. Microbe-Mediated Synthesis of NPs

Compared to conventional approaches, the utilization of microbial machinery for the biosynthesis of NPs offers several advantages, including enhanced efficiency, improved safety, and greater environmental sustainability [68]. In recent studies, researchers have investigated the potential of diverse microorganisms to produce NPs. Such types of NPs include metals, metal oxides, and other materials that can be synthesized via both intracellular and extracellular mechanisms [69]. The use of *Saccharomyces cerevisiae* and *Cryptococcus humicola* for the production of magnetically responsive FeO NPs has been documented. Nadeem et al. (2021) studied the use of different types of molds, yeasts, and bacteria to produce FeO NPs. Reports have suggested that microbes are among the best sources for synthesis, and they have recently been used to produce a number of metallic NPs, especially iron NPs [70]. The use of fungi for the synthesis of NPs is characterized by enhanced efficiency and cost-effectiveness compared to their bacterial counterparts, primarily because of the heightened propensity for metal accumulation in fungi. Furthermore, the use of biomass and subsequent processing are more straightforward for the synthesis of NPs using fungi. Khalil et al. (2022) prepared Fe NPs using a green approach and described their inherent potential in the microbiological realm. Biosynthesis can be catalyzed by biomolecules and enzymes found in microbial cells. These microbial-generated inorganic NPs have the potential as agents for cancer therapy [71]. Jadhav et al. (2022) synthesized Fe-based NPs with paramagnetic and ferromagnetic properties. This synthesis was aimed at augmenting microbial activity and mitigating the inhibitory effects on methanogenesis [72]. The synthesis of the biological-based Fe NPs is shown in Figure 11.

## 3. Synthetic and Structural Insights of Fe NPs

Iron nanoparticles (INPs), commonly referred to as Fe NPs, have garnered significant attention owing to their fascinating properties and wide range of applications in various fields, such as materials research, environmental remediation, and medical care. These nanomaterials have captivated the scientific community and have been the subject of extensive investigations [73]. This study highlights the remarkable characteristics of Fe NPs and their diverse applications, highlighting their potential impact in advancing scientific knowledge and addressing societal challenges. To optimize the potential of Fe NPs, it is imperative to comprehensively understand their synthesis and functionality. Table 1 presents the synthetic strategies, biological evaluations, and structural properties of Fe NPs.

## 4. Factors Influencing the Size, Shape, and Surface Properties of Fe NPs

Understanding the synthetic methodologies and structural properties of Fe NPs is essential for harnessing their full potential [8]. These insights are pivotal for tailoring Fe NPs to specific applications and optimizing their performance in diverse contexts [30]. Silica-coated iron oxide nanoparticles (SiO Fe NPs) have garnered significant interest in the biomedical field, particularly in diagnosis and therapy, owing to their distinctive characteristics including superparamagnetism, facile surface functionalization, and colloidal stability [88]. The key factors that influence the properties of the Fe NPs are depicted in Figure 12.

The impact of the particle size on the human body was significant. NPs with a hydrodynamic diameter of less than 100 nm exhibit superior delivery effectiveness compared with their larger counterparts [89]. Additionally, it has been observed that nearly neutrally charged inorganic Fe NPs, characterized by zeta potential values ranging from (−10 to +10 Mv), exhibit superior delivery efficiencies compared to INPs with larger zeta potential values. Moreover, it has been observed that rod-shaped inorganic INPs exhibit higher efficacy in drug delivery applications compared to their spherical or flake-like counterparts [90]. The synthesis of pure hematite NPs influences both the size and morphology. Additionally, it was found that the concentration of the precursor not only affects the particle size, but also affects the crystallinity, development of morphology, and formation of lumps in the end product [91]. Figure 13 depicts the factors that affect the surface properties of Fe NPs.

The amount of plant extract exerted a notable influence on the particle size of FeO NPs manufactured using green methods. For example, the augmentation of tangerine peel extract concentration from 2% to 6% yielded a noteworthy decrease in the size of the FeO NPs, from 200 nm to 50 nm. Nevertheless, a higher concentration of up to 10% results in the agglomeration of NPs, characterized by their strong attachment to one another [92]. The FeO NPs prepared using *H. vulgare* leaf extract exhibited colloidal instability and an aggregation propensity. In contrast, the particles generated from *R. acetosa* extracts exhibited a notable degree of stability [91,93]. NPs with spherical surface shapes and crystal structures are preferred over their amorphous counterparts. Longer periods of elevated temperatures throughout the synthesis may account for the observed surface morphology shift from spherical to rod-like [39].

## 5. Characterization Techniques for Analyzing the Physiochemical Properties of Fe NPs

Various microscopy-based techniques are available to provide comprehensive information regarding the dimensions, morphology, and crystalline structure of NPs [94]. Characterization of the physicochemical properties of Fe NPs is shown in Figure 14.

### 5.1. X-ray-Based Techniques (XRD)

X-ray diffraction (XRD) is a commonly employed technique for the characterization of NPs, enabling the determination of their chemical composition, crystalline structure, and phase identification. The broadening of the XRD diffraction lines is influenced by the size of individual particles, which manifests as peaks in the recorded data [95]. The Debye–Scherrer equation was employed to determine the typical size of the INPs crystals.
D = kλ/βCosθ

In this equation, “D” denotes the average size of the material in nanometers, “k” represents the Scherrer constant with a value ranging from 0.9 to 1, “λ” signifies the wavelength of X-rays, and “β” represents the Scherrer constant with a value maximum [96]. A report has shown that the XRD pattern obtained from the INPs exhibited distinct peaks at 2θ values of 24.16°, 30.4°, 33.24°, 35.8°, 43.5°, 54.1°, and 57.4°. These peaks can be attributed to the (012), (104), (200), (311), (511), and (440) crystallographic planes, respectively [95]. Kayani et al. (2014) revealed that FeO is composed of a hematite phase, as determined by X-ray powder diffraction analysis. NPs’ average crystalline size grew from 34 nm to 36.7 nm as the annealing temperature was raised from 400 °C to 1000 °C [97]. Ashraf et al. (2023) collected evidence from XRD patterns of the crystalline structure of FeO NPs. Bragg reflection peaks were measured at 2θ = 24, 33, 35, 40, 49, and 53. Using the Scherrer equation, the mean size was determined to be 2.45 nm [98]. Kamal et al. (2023) synthesized FeO NPs, which exhibited the following XRD pattern: the diffraction patterns of the manufactured NPs were recorded from 10 to 70°. The diffraction peaks of the produced material had a conventional structure with many peaks representing iron at 2θ = 30.26°, 32.33°, 34.67°, 36.06°, 43.47°, and 46.36° [99].

### 5.2. UV–Visible Spectroscopy

This technique was employed to monitor the progress of the NP production process. Reports have shown that the spectral data obtained for the produced Fe NPs employing different leaf extracts exhibited a range of 300–500 nm, which corresponds to the spectral properties of metallic iron [100,101]. The instrument uses incident light to pass through a specimen, whereas a detector positioned on the opposing side measures the intensity of the transmitted light. It is recognized as a highly dependable method for the investigation of NP colloids, even when applied in industrial settings [102]. Afrouz et al. (2023) prepared Fe_3_O_4_ NPs using spinach extracts and reported their UV/Vis spectra. A significant peak in the range 350–400 nm demonstrated that Fe_3_O_4_ was produced and was stable. Furthermore, the absorption peak confirmed the nano-scale production of the Fe_3_O_4_ NPs. According to UV/Vis measurements, spermine showed no detectable peaks. Additionally, UV/Vis spectra demonstrated that spermine binding on the surface of Fe_3_O_4_@SiO NP caused a dramatic drop in the peak at 350–400 nm [103]. Iqbal et al. (2020) used UV–visible spectroscopy to characterize FeO NPs under aquatic conditions. This was made possible by passing the solution through a wavelength range of 200–800 nm. Maximum absorption was observed at 341 nm. The first point was a protrusion, whose peak was attributed to the surface plasmon resonance of FeO NPs which was discovered to be around [104]. Lohrasbi and his colleagues (2019) have demonstrated the green synthesis of Fe NPs that prefabricated nanosheets had shown the ability of removing methyl orange (MO) contamination. The MO concentration during fabrication was assayed using UV-Vis spectroscopy supported with Fe-catalyzed H_2_O_2_. It can be concluded from the study outcome that the mixture of FeO NPs and H_2_O_2_ has 83.33% of dye decolorization efficiency within 6 h [105].

### 5.3. Dynamic Light Scattering (DLS)

DLS is a commonly used non-destructive method of spectrum analysis with electromagnetic radiation that is capable of determining both materials and particle size in aqueous media [106]. The nature of the particle size seems to hold paramount importance for the optimization and synthesis of NPs. The particle size assessment by the DLS method gives the hydrodynamic diameter of the synthesized NPs, whereas the TEM and powdered XRD-based particle size analysis showed smaller particle sizes than estimated by DLS. This phenomenon is attributed to the quantification of the diameter of the sphere encompassing the NPs [107]. This approach is additionally applied in the area of NPs for the assessment of morphology and dimension description [100,108]. Aksu et al. (2019) used *Ficus caricato* to prepare FeO NPs. The recorded diameter of the FeO NPs was found to be 475 nm using DLS analysis, which gave an average value based on the intensity [109]. Harmansah et al. (2022) synthesized Fe_3_O_4_ NPs using banana peel extract, and DLS calculations revealed that the average particle size distribution was between 44 and 58 nm. Owing to this aggregation, the findings were found to be more than anticipated. MNPs (St Dev), with an average size of approximately 90 nm, were synthesized in a green environment. MNPs had a PDI of 0.446, and DLS measurements showed that most particles were 924.5 nm in size. When the PDI value was less than 0.7, the molecules were evenly distributed throughout the green-synthesized FeO NPs [110]. Hassan et al. (2022) synthesized FeO NPs using the seed extract derived from *Nigella sativa* and found that the mean diameter of the produced NPs was approximately 31.45 nm [111].

### 5.4. Fourier-Transform Infrared Spectroscopy (FT-IR)

Because of its nature as a vibrational spectroscopy, FTIR can serve to determine which of the NP functional groups are responsible for a given IR absorption band [112]. The utilization of FTIR aids in the acquisition of structural information of NPs through the analysis of vibrational modes associated with the bonds found inside Fe NPs [113]. Stoia et al. (2016) employed FTIR spectroscopy and proved that there was the production of a mixture of maghemite and magnetite. The spectra of the FTIR of magnetite exhibited two prominent IR absorption bands at 570 and 390 cm^−1^. Maghemite, being a modification of magnetite that possesses structural irregularities, has absorption bands at different wavenumbers of 630, 590, and 430 cm^−1^ [114]. Alangari et al. (2022) performed an FTIR study on Fe_2_O_3_ NPs and revealed a spectral range of wavenumbers between 400 and 4000 cm^−1^. This analysis provided constructive information for identifying the chemical bonds and functional groups present in Fe_2_O_3_ NPs. The FTIR spectrum of the manufactured Fe_2_O_3_ NPs displayed a discrete set of absorption bands at various wavenumbers of 517, 621, 1020, 1612, and 3435 cm^−1^. The peaks at 517 cm^−1^ and 621 cm^−1^ were attributed to the inherent vibrational stretching of the Fe-O bond. These observations indicated that the NPs were composed of FeO [115]. Zakariya et al. (2022) explored the functional groups of FeO NPs using FTIR. A suspension of the FeO NPs was deposited on the sample holder. The transmittance spectra of the samples were measured in the wavelength range of 400–4000 cm^−1^ with a resolution of 4 cm^−1^ [116].

### 5.5. Scanning Electron Microscopy (SEM)—Energy Dispersive X-ray Spectroscopy (EDX)

SEM is a highly prevalent instrumental technique employed for the examination and analysis of micro- and NP-imaging characterization of solid materials. This is considered a very efficient approach for examining both organic and inorganic substances [117]. Kiwumulo et al. (2022) conducted SEM and revealed that FeO NPs had a granular, homogeneous, spherical morphology, characterized by a mean diameter of approximately 16 nm. In addition to confirming the formation of FeO NPs, EDX was employed to obtain further structural information regarding the solution. The EDX spectra exhibit Fe peaks in three distinct regions: 0.66, 0.68, and 7.0 [118]. Hassan et al. (2022) produced green-synthesized FeO NPs with an average diameter of 31.45 nm, which were observed using SEM to illustrate the dimensions and morphology of the mostly spherical NPs [111]. Haris et al. (2023) investigated the geometry of biosynthesized FeO NPs using SEM and confirmed the presence of trigonal rhombohedral crystalline forms, which is consistent with the XRD findings. The formation of distinct shapes in large particles can be attributed to the crystal development process. The examination conducted using EDX analysis provided qualitative and quantitative information regarding the presence and concentration of Fe components involved in the production of FeONP. Absorption peaks were observed in the energy range of 6–7 keV as a result of the surface plasmon resonance exhibited by Fe NPs [119].

## 6. Antifungal Mechanism of Fe NPs

Several recent reports have summarized antifungal mechanisms [120,121], fungal infections of Fe NPs in humans [122], and the effect of Fe NPs against bacteria [123]. The researchers synthesized IONPs by utilizing an environmentally friendly methodology and subsequently investigated their antifungal activities against *Aspergillus niger* and *Mucor piriformis* [82]. The authors explained that the interaction between INPs and fungal proteins results in the deactivation of certain proteins and their direct contact with DNA. Consequently, the process of interaction leads to mutations that hinder DNA replication [124]. Furthermore, the small dimensions of NPs facilitate their ability to readily traverse cellular membranes. The accumulation of substances in the cellular membrane has the potential to trigger cell rupture, a process known as cell lysis [87]. Moreover, it has been observed that NPs exhibited the capability to permeate fungal spores by traversing the cell membrane. This penetration results in the emergence of an electron-light area located near the nucleus of the cell. Subsequently, particles disrupt the respiratory pattern, ultimately resulting in the complete inhibition of cell division, thereby inducing cellular demise [125,126,127]. In human pathogenic yeasts, antifungal drugs have been shown to cause intracellular buildup of reactive oxygen species (ROS) [128]. ROS have demonstrated potential as highly effective tools for antifungal therapy owing to the oxidative damage that ensues from their accumulation [129]. The antifungal activity of Fe NPs is mediated by a direct connection between the NPs and the surface of fungal cells. This interaction affects the permeability of cell membranes and facilitates the entry of NPs into the cells. Consequently, the presence of NPs induces oxidative stress within fungal cells, leading to the inhibition of cell development and, ultimately, cell death [82]. The antifungal efficacy of Fe NPs is contingent on their chemical composition as well as their physical attributes, including form, size, solubility, agglomeration, and charge at the surface [130]. Particle size is a critical factor in the assessment of the antimicrobial efficacy of INPs as it influences their ability to penetrate the cell walls of microorganisms via carrier proteins or ion channels. Consequently, a reduction in particle size leads to the enhanced internalization of INPs into microbial cells, leading to the suppression of DNA or RNA production [129,131].

The responses and genetic damage of NPs inside a system are determined by the agglomeration process [132]. Nevertheless, when particles are not organized into agglomerates, they can disperse themselves, thereby enhancing the formation of reactive oxygen species (ROS) formation [133]. In addition, the surface charge of the material plays a crucial role. To modify the surface potential and introduce functional groups, INPs were coated with a chitosan polymer [134]. Owing to these mechanisms, chitosan has the potential to effectively restrict microbial growth through processes such as chelation of transition metallic ions, enzyme inhibition, and disruption of exchange with the surrounding media [76]. The high zeta potential of INPs facilitates robust interactions, leading to membrane rupture and decreased viability [135].

## 7. Optimization Strategies for Enhancing Antifungal Properties

Significant progress has been made in the development and implementation of many strategies aimed at addressing clinical obstacles and augmenting the therapeutic efficacy of membrane-active antimicrobials [136]. Several approaches have been implemented to enhance the application capabilities of Fe NPs. These include surface modification, implementation of protective shells, utilization of solid supports, and the introduction of a second metal via doping [137]. The antifungal effects of both uncoated and coated FeO NPs were evaluated against the *Trichoderma fungus*. The experimental findings have indicated that the FeO NPs coated with PAH had a more potent antifungal effect than uncoated IO NPs [138]. FeO NPs can be modified by functionalization or coating with synthetic or natural polymers such as chitosan, organic surfactants, inorganic chemicals, or bioactive compounds. This modification aimed to improve the stability, biological compatibility, and antimicrobial activity of NPs [139]. Recent advancements have led to enhanced functional capabilities through the utilization of hybrid IO NPs, including Ag–IO, Co–IO, Au–IO, Mn–z IO, and Cu–IO. IO NPs with sizes ranging from 10 to 30 nm have been synthesized [140]. The observed FeO NPs exhibited potent antifungal properties against five distinct fungal strains. The application of chitosan for coating Fe NPs results in increased ROS generation, thereby leading to an increase in antifungal efficacy [141,142]. Figure 15 shows various types of optimization strategies for enhancing antifungal properties.

## 8. Antifungal, Biocompatibility and Toxicity Considerations

In recent years, the field of using NPs as potential antifungals has seen significant progress, particularly in the realm of biomedical applications, where Fe NPs have emerged as a noteworthy and rapidly growing area of interest. Furthermore, the ability of a material to interact with live tissues without producing any reactions is known as biocompatibility [70], and to this end, numerous contributions have been presented by researchers around the globe. However, toxicity considerations entail the assessment of how a material affects living organisms. Determination of dosage, exposure routes, and possible long-term effects are all part of this process [143]. Safety and medication factors are important for guaranteeing the efficacy of Fe NPs [144]. The following section describes the antifungal potential, biocompatibility, and toxicity considerations of Fe NPs.

### 8.1. Antifungal Activity of Fe NPs

The scientific community confronts a formidable challenge in dealing with the emergence of diverse species of fungi that exhibit resistance to traditional antimycotics. To address this issue, the development of novel antifungal agents through the integration of nanotechnology represents a burgeoning field of investigation. One potential solution entails the utilization of NPs or the incorporation of NPs into formulations as an antifungal agent. Prucek et al. (2011) reported two types of magnetic binary nanocomposites, Ag@Fe_3_O_4_ and γ-Fe_2_O_3_@Ag. These nano-composites were synthesized using a chemical reduction approach, and both systems exhibited significant antifungal activity against ten bacterial strains. The results indicated that the MIC ranged from 15.7 mg/L to 124.9 mg/L, while the MTT assay for cytotoxicity evaluation against mouse embryonic fibroblasts showed 430 mg/L and 292 mg/L for Ag@Fe_3_O_4_ and γ-Fe_2_O_3_@Ag, respectively [145]. Niemirowicz et al. (2016) carried out a study to assess the antifungal potential, biofilm inhibition activity, and hemolytic characteristics of two commercially available antibiotics that were anchored to the surface of Fe NPs. The study involved examining the developed model against the isolation of pathogens from clinical cultures of *Candida species* and human RBCs. The nanoscale systems composed of Fe NPs and antibiotics demonstrated stronger fungicidal potential than their unbound counterparts. Additionally, the researchers detected a synergistic effect upon combining antibiotics and Fe NPs against all *Candida* strains used in the study. Results showed that nano-systems were more effective than unbound agents in combating *Candida* strains in the presence of pus, and they demonstrated the ability to prevent *Candida* biofilm formation as an antifungal agent. Furthermore, improvement in their biocompatibility has also been demonstrated [146]. According to Seddighi et al. (2017), the antifungal activity of Fe NPs against different species of *Candida* was assessed and compared with a drug (fluconazole) that is frequently used in clinical settings. The researchers manufactured spherical Fe NPs with a size ranging from 30 to 40 nm. The MIC and MFC of the FLZ values were within the range of 16–128 µg/mL and 64–512 µg/mL, respectively. However, the MIC and MFC values of Fe NPs ranged from 62.5 to 500 µg/mL and 500 to 1000 µg/mL, respectively. *Candida tropicalis*, *Candida albicans*, and *Candida glabrata* showed the highest susceptibility to Fe NPs in the growth inhibition evaluation. The obtained NPs showed antifungal properties against pathogenic *Candida* species, and thus exhibited inhibition growth for all the examined *Candida* species [147]. According to Salari et al. (2018), the biofilm-producing ability of the *Candida* strains was evaluated as well as the *Candida* strain biofilm inhibition by Fe NPs, which was also evaluated and compared with FLZ following the MTT assay. It was observed that the biofilm-creating potential of *C. lusitaniae* was increased by much as compared to other examined *Candida* strains. However, a significant level of biofilms was achieved by all the tested *Candida* strains. Furthermore, the biofilm reduction effect of fluconazole for *C. tropicalis*, *C. krusei*, and *C. lusitaniae* was statistically greater than that of Fe NPs [148]. As described in Table 1, Nehra et al. (2018) reported on their research findings regarding the synthesis of pristine and chitosan-coated Fe NPs and demonstrated its antifungal activity against multiple strains (76). Parveen et al. (2018) used a phyto-mediated approach to synthesize Fe NPs (10–30 nm), with tannic acid serving as a capping and reducing agent. The authors subsequently evaluated the antifungal activity of these Fe NPs against five diverse types of fungal pathogens. The results indicated that Fe NPs exhibited considerable antifungal potential against all examined fungal pathogens, with *T. roseum* (87.73%) displaying the highest inhibition followed by *C. herbarum* (84.88%). The maximum zone of inhibition caused by Fe NPs was observed against *P. chrysogenum* (28.68 mm), while *P. chrysogenum* also displayed the highest activity index of 0.81. Notably, MIC values of Fe NPs varied from 0.063 to 0.016 mg/mL for the distinct kinds of fungal pathogens and were comparable to that of the standard, which ranged from 0.004 to 0.016 mg/mL. The study demonstrated the efficacy of Fe NPs against a variety of fungal pathogens [75].

A recent study by Devi et al. (2019) on the phyto-mediated synthesis of Fe NPs using *P. orientalis* leaf extract, has shown promising results in the antifungal activity against two test strains, *Aspergillus niger*, and *Mucor piriformis*. According to the findings, Fe NPs (size 38 nm, α-Fe_2_O_3_, and γ-Fe_2_O_3_) displayed significant antifungal activity against both species, acting as effective model fungi. It is noteworthy that the antifungal activity of Fe NPs was more pronounced against *M. piriformis* [82]. De Lima et al. (2020) developed a nano-system by loading FLZ onto chitosan-coated Fe NPs and assessed the biofilm development potential of *Candida albicans* and *Candida glabrata* strains. They also compared the antibiofilm effects of Fe NPs with those of FLZ. The results of the study revealed that the chitosan-coated Fe NP-FLZ nano-system exhibited greater efficacy in lowering MIC values. Moreover, it demonstrated effectiveness in suppressing planktonic *Candida* cells when compared to FLZ utilized alone [149]. A report published by Zare-Khafri et al. (2020) showed the antifungal properties of Fe NPs against FLZ-resistant *Candida albicans* flora. The authors performed an estimation of *ERG11* gene expression levels as well as an estimation of ergosterol proportion. This study shows that 93% of *C. albicans* isolates were found to be resistant to FLZ; however, those magnetic Fe NPs revealed antifungal properties towards FLZ-resistant colonizing species with MICs between 250 and 500 µg/mL. Also, it was observed that the expression of the *ERG11* gene was downregulated in the FLZ-resistant colonizing separates of *C. albicans* and that the ergosterol concentration was lowered [150]. Sriramulu et al. 2020 used the green approach of biosynthesis with *Aegle marmelos* extract and the final step was the calcination process at 400 °C. Moreover, the authors compared the effect of the Fe NPs against soil-isolated pathogenic plant fungi *F. solani* (12 ± 0.53 mm) with FLZ (7 ± 0.38 mm) at 30 µg/mL *F. solani* [151]. Similarly, Win et al. (2021) reported on their research findings regarding the synthesis of Fe NPs through the use of an aqueous extract obtained from the green *microalga Chlorella* K01 and demonstrated its antifungal activity against multiple strains, as shown in Table 1 [81]. Researchers from a recent study, Yasin et al. (2023), utilized an environmentally friendly technique to produce Fe NPs. This green method involved the use of an aqueous extract of *Laurus nobilis* leaves. They evaluated the antifungal properties of these Fe NPs, as well as the commercially available metalaxyl–mancozeb antifungal agent against *A. alternata* strains. The Fe NPs produced through the green approach showed increased antifungal activity against two strains (*A. alternata* OR236467 and *A. alternata* OR236468) at a concentration of 800 ppm, with relative GI proportions of 75.90% and 60.64%, respectively. This study highlighted the potential of the biogenic Fe NPs derived from fresh leaves of *L. nobilis* to serve as biofungicides [152]. Recently, Azadi et al. (2024) have utilized the catalytic properties of Fe_3_O_4_@SiO_2_/Schiff-base/Cu NPs (a nano-antifungal agent) to evaluate its antifungal efficacy and cytotoxicity and describe the potential use of Cu NPs in the arena of fungicides when embedded with Fe NPs. The study specifically targeted the antifungal capabilities of Fe_3_O_4_@SiO_2_/Schiff-base/Cu(II) nanoparticles against multiple strains [153].

### 8.2. Assessment of the Biocompatibility of Fe NPs

Given the potential benefits of NPs in a wide range of healthcare applications, evaluating the biocompatibility of Fe NPs for biomedical applications is an important field of research [154]. Fe NPs have attracted considerable attention owing to their unique chemical and physical properties. These qualities make them attractive candidates for use in targeted cancer therapies, as bio-imaging agents, and in drug delivery systems [155]. The biocompatibility of Fe NPs depends on several factors that must be considered. During an assessment process, it is recommended that the fact that products are not toxic to living organisms is essential. For toxicity studies, the Fe NPs cytotoxicity is ascertained by exposing them to different cell lines and animal models [156,157]. Figure 16 demonstrates the key biocompatibility characteristics of Fe NPs.

After the analysis of toxicity, the next usual step is analyzing the interactions between biological systems and Fe NPs. This is also a crucial step in the accomplishment of the biocompatibility of Fe NPs. This determines the examination of the immunological system’s inability to recognize Fe NPs, their potential to cause an inflammatory response, and their stability [158]. Advanced techniques, such as electron microscopy, molecular imaging, and spectroscopy have been employed to acquire more about how Fe NPs develop interactions with biological systems [159]. Such approaches provide vital indications concerning the pattern of NPs distribution, cellular uptake process, and the subsequent changes that occur in cells and their subcellular structures [160].

### 8.3. Assessment of Toxicity of Fe NPs and Strategies to Mitigate Risks

Several key considerations related to the assessment of toxicity and biocompatibility are of high significance for the suitability and utilizing of NPs, understanding their possible toxicity, and developing strategies to reduce hazards [161]. The special qualities and prospective applications of Fe NPs have garnered much interest in several areas, including the environment and medicine. To guarantee their safe usage, their potential toxicity must be carefully evaluated [162]. Fe NPs’ toxicity is caused by many factors including shape, surface coating, size, and reactive oxygen species (ROS) production [163].

The potential of Fe NPs to cause oxidative stress and subsequent cellular damage is one of the main causes of concern and is of paramount interest as well. If such factors are not taken into consideration, they might impart DNA damage, inflammation, and eventually cell death. Fe NPs can produce ROS via Fenton or Haber–Weiss reactions, resulting in the formation of highly reactive -OH radicals [164]. As a result, approaches for reducing toxicity levels must focus on reducing ROS generation and governing their interaction with biological components [165]. Surface modification of Fe NPs can be used to improve biocompatibility and reduce toxicity. Fe NPs can be coated with biocompatible substances, including biomolecules or polymers, to create a barrier that reduces their direct contact with biological components [166]. Surface changes can also improve solubility, dispersibility, and stability, making them more appropriate for use in biomedical applications. Optimizing the size and form of Fe NPs is another way to reduce the associated hazards connected with them [167]. Furthermore, modifying the shape of NPs can affect their dispersion inside the cells, their cellular absorption, and possible toxicity [167]. Toxicological evaluations, both in vivo and in vitro, are essential to determine whether Fe NPs are safe. Studies conducted on cell cultures have shed light on genotoxicity, cytotoxicity, and the underlying mechanisms of toxicity [168]. The toxicity evaluation of the Fe NPs is shown in Figure 17.

The effects of Fe NPs on different tissues, organs, and physiological systems have been further clarified by animal investigations. Through the implementation of these thorough assessments, scientists can recognize the negative consequences and create suitable plans to reduce risks [169]. Understanding the behavior of Fe NPs in various environmental compartments, such as soil, air, and water, is necessary to determine their overall toxicity and environmental impact [170]. In conclusion, determining the biocompatibility and compatibility of Fe NPs is essential for their use in a responsible and secure manner [171].

### 8.4. In Vitro and In Vivo Studies on the Biocompatibility and Toxicity of Fe NPs

Understanding the possible uses of Fe NPs in various industries, including biomedicine, depends heavily on in vitro and in vivo investigations of the toxicity and biocompatibility of these nanomaterials [172]. Fe NPs exhibit excellent characteristics, making them a good option for therapeutic, imaging, and drug-administration applications. However, it is crucial to evaluate their toxicity profiles and biocompatibility before these applications can be implemented [173]. Although in vivo research has examined the impact of Fe NPs on living organisms, usually animals, in vitro investigations have been conducted in a controlled environment without the presence of a living organism [160,174]. Different cell lines were tested at varying NP dosages to observe how cells respond to oxidative stress, proliferation, and viability [175]. These studies allow for the measurement of cellular uptake and intracellular distribution, and the subsequent effects on cellular functions. For instance, researchers have investigated the impact of Fe NPs on gene expression, biological signaling networks, and protein synthesis [176]. A variety of techniques have been used to assess biocompatibility in vitro, including trypan blue exclusion, MTT, and LDH cytotoxicity assays. Additionally, methods like confocal microscopy and flow cytometry allow for a thorough examination of the interactions between cells and Fe NPs [177]. A study has revealed that optimally synthesized Fe NPs typically exhibit lower toxicity and better biocompatibility [178].

Animal models have been used to study the possible systemic toxicity and long-term effects of exposure to Fe NPs. In these investigations, NPs were administered via oral, intravenous, or intraperitoneal routes of exposure, and their effects on numerous organs, hematological parameters, immunological responses, and general health were evaluated [179]. Several advanced imaging techniques, such as computed tomography (CT), positron emission tomography (PET), and magnetic resonance imaging (MRI), are useful in tracing the accumulation, distribution, and elimination of Fe NPs within living organisms. [180]. By examining biodistribution patterns and tracking possible changes in organ function, scientists can learn more about the general safety profile of Fe NPs. Histopathological examination of the treated tissues also sheds light on possible immunological reactions, inflammation, and tissue damage caused by Fe NPs [159,181]. To improve the accuracy and repeatability of these investigations, variables such as dosage dependence, aggregation state, surface modification, and particle size must be properly considered and standardized [182].

## 9. Applications of Optimized Fe NPs in Biomedicine

In biomedicine, the researchers’ main goal is improving Fe NPs properties by developing their unique characteristics and diverse applications [183]. The field of biomedicine has shown great interest in optimizing Fe NPs because of their special qualities and possible uses. NPs have demonstrated encouraging results in a range of biomedical applications and have outstanding biocompatibility. In this section, we examine some of the most important uses of optimized Fe NPs in biomedicine [184]. Targeted medication delivery is one of the main areas in which optimized Fe NPs are used. These Fe NPs can be functionalized with targeting ligands and encapsulated with certain medications to transport them directly to the intended location within the body [185]. The small size and wide surface area of Fe NPs allow for better cellular absorption and effective passage across biological barriers, which improves drug delivery [186]. Figure 18 shows key areas and the applications of Fe NPs in biomedicine.

Furthermore, tuned Fe NPs have become useful for medical imaging. These NPs can be utilized as contrast agents in MRI, owing to their superparamagnetic characteristics [187]. They provide high-resolution images and significant signal intensities, making it possible to diagnose a variety of disorders accurately. Moreover, Fe NPs can be combined with other imaging modalities, such as fluorescent labels or radioisotopes, to facilitate multimodal imaging and enhance diagnostic precision, such as fluorescent labels or radioisotopes [188]. An interesting additional application of the enhanced Fe NPs is in heat shock-based cancer therapy. Fe NPs can specifically destroy cancer cells while shielding healthy tissues and can be used as a tumor treatment technique. Fe NPs can also be engineered to obtain specific shapes, surface properties, and sizes, to improve their therapeutic outcomes and optimize their heating efficiency [189].

### 9.1. Antifungal Coatings for Medical Devices and implants

Recently, there has been considerable increasing interest in employing Fe NPs because of their unique qualities and use in several industries, including medicine. Their application in antifungal coatings for implants and medical equipment has yielded encouraging outcomes [190]. Patients with implants or other medical devices are at considerable risk of fungal infections, which can result in life-threatening problems and prolonged hospital admissions. Conventional antifungal therapies, including oral drugs and topical creams, may be insufficient to stop or cure these infections. However, adding Fe NPs to implants and medical device coatings may provide a unique remedy [140]. Fe NPs allow for the regulated release of Fe ions, which is one of their key benefits. The growth of numerous fungal species is inhibited by Fe ions, which are well known for their antifungal properties [191]. Through the release of these ions, coated implants and medical devices efficiently prevent infections by creating an environment averse to fungal development and colonization [192]. Moreover, Fe NPs have outstanding biocompatibility, meaning that living tissues may accept them without experiencing major toxicity or negative side effects. This is an important consideration for guaranteeing the safety and effectiveness of antifungal coatings, as any possible toxicity could endanger the patient and jeopardize the operation of the implant or medical device [193].

Studies have shown that coatings based on Fe NPs can prevent fungal infections. In vitro research has demonstrated that these coatings have potent antifungal properties against a broad spectrum of fungal species, such as *Aspergillus fumigatus* and *Candida albicans*, which are frequently occurring pathogens linked to illnesses related to medical devices [194]. Furthermore, compared to untreated surfaces, in vivo investigations using animal models have demonstrated that surfaces coated with Fe NPs considerably reduce fungal cell colonization and proliferation [195]. When applied to medical device coatings, Fe NPs offer additional advantages beyond their antifungal characteristics. To ensure the lifespan and resilience of coatings to wear and tear, they can improve their mechanical strength and durability [191]. To generate multifunctional coatings that provide broader protection against various pathogens, Fe NPs can be functionalized with other medicinal or antibacterial substances [59].

### 9.2. Fe NPs as Targeted Drug Delivery Systems for Antifungal Agents

Fe NPs have attracted considerable interest as possible targeted drug delivery vehicles for antifungal drugs [196]. Fe NPs have special physicochemical characteristics that make them a good option for enhancing antifungal therapy, such as their small size and large surface area [197]. The advantages and difficulties of employing Fe NPs as medication carriers to treat fungal infections are discussed in this section [198]. The capacity of Fe NPs to encapsulate and transport antifungal drugs directly to the site of infection is one of their main benefits. The small size of Fe NPs enables them to pass through the fungal cell wall and enter intracellular compartments, thereby improving the effectiveness of drug delivery [199]. Moreover, targeting ligands, which attach exclusively to fungal cells, can be added to the surface of Fe NPs to modify their targeting properties, thereby increasing the efficiency and selectivity of drug delivery. By minimizing drug exposure in healthy tissues, this targeting method lowers the likelihood of side effects [200].

Fe NPs can be used in biological applications because of their exceptional stability and biocompatibility. They can shield encapsulated medications from degradation and are themselves less prone to it [201,202]. By enabling extended drug delivery, this controlled-release feature reduces the number of administrations required and encourages patient compliance. The use of Fe NPs for antifungal therapy is fraught with challenges. The possible toxicity of Fe NPs is an issue [203]. In vitro studies demonstrated that elevated levels of Fe NPs can cause cellular damage by producing ROS. To reduce toxicity and increase therapeutic efficacy, rigorous optimization of the NP size, surface coating, and drug-loading capacity is required. Furthermore, in vivo investigations are essential to assess the long-term consequences and biocompatibility of Fe NPs, accounting for elements such as biodistribution, clearance, and possible build-up in critical organs [3].

### 9.3. Fe NPs in Wound Healing and Tissue Regeneration

The study of tissue regeneration and wound healing has been of great interest to Fe NPs because of their unique qualities and possible therapeutic uses. Fe NPs are adaptable to a range of biological applications because of their high biocompatibility and ease of synthesis in various forms and sizes [204]. Fe NPs exhibit excellent antibacterial properties and can effectively prevent and treat infections commonly associated with wounds. Fe NPs possess antimicrobial activity by generating ROS that target and destroy pathogenic bacteria, while minimizing damage to healthy tissues [205]. Figure 19 shows the effect of the Fe NPs in wound healing.

Fe NPs also promote tissue regeneration by triggering important cellular functions. Owing to their antioxidant properties, they lower oxidative stress in the wound environment and create an ideal setting for tissue healing. Fe NPs also exhibit anti-inflammatory properties by regulating the expression of inflammatory cytokines. This reduces excessive inflammation, which can impede wound healing [206]. Fe NPs also improve collagen synthesis, which is essential for wound healing. Fibroblasts, cells that produce collagen, are activated by Fe NPs, which accelerates wound closure and improves tissue regeneration. Research has demonstrated that Fe NPs can upregulate the expression of growth factors essential for angiogenesis and tissue regeneration, including transforming growth factor-beta (TGF-β) and vascular endothelial growth factor (VEGF) [207]. Fe NPs can be added to a variety of scaffolds and wound dressings to continuously release Fe ions, thereby enhancing their therapeutic effects. Treatment at the site of injury can be customized and localized for treatment because of the controlled-release kinetics that these nanocomposite materials can be made to have [208]. The tissue regeneration process using Fe NPs is shown in Figure 20.

It is imperative to acknowledge that a comprehensive assessment of the toxicity and biocompatibility of Fe NPs must be conducted before their clinical implementation. Studies conducted in vitro and in vivo are essential to evaluate the safety profile of Fe NPs, including their immunological response, cytotoxicity, genotoxicity, and hemocompatibility [209,210].

### 9.4. Fe NPs in Cancer Therapy and Diagnosis

Global Cancer Observatory data indicate that cancer problems are among society’s worst scourges, with 45% of cancer patients dying from the disease. Lung and breast cancer have the highest estimated mortality rates [211]. The field of cancer nanomedicine has witnessed the development of numerous products that have greatly enhanced cancer care, owing to extensive research and substantial preclinical investment [212]. However, there is a prevailing perception that cancer nanomedicine has not fully delivered on its initial promise, largely due to the limited number of licensed drugs and their clinical performance [213]. Magnetic NPs have emerged as a versatile biomedical platform, offering a wide range of applications and controllable parameters that significantly impact the treatment’s effectiveness [214,215]. Fe NPs have a long history of clinical usage, and their use has produced a wide range of therapeutic products [216,217]. While conventional treatments, such as surgery for solid tumors, antitumor drugs, and radiation, have undoubtedly been effective in preserving lives, it is important to acknowledge the significant adverse side effects they can cause [218,219]. In recent years, there has been a growing interest in the application of magnetic NPs in the field of biomedicine [220,221]. They emerged as a novel therapeutic option in the field of magnetic resonance imaging and cancer treatment [218]. Yang et al. (2023) developed multilayer Fe_2_O_3_ architectures (UF@PPDF NPs) by modifying ultra-small γ- Fe_2_O_3_ nanocrystal assemblies and conjugating cancer cell-targeting folic acid (FA) with the chemotherapeutic medication doxorubicin (Dox). UF@PPDF NPs demonstrate very effective photothermal treatment (PTT) following NIR-II irradiation, as well as the regulated release of ferric ions and Dox, resulting in a trifunctional synergistic cancer therapy for tumor ablation [222]. Chen et al. (2021) developed a nanosystem coated with ferrous ferric oxide (Fe_3_O_4_) and chlorin E6 (Ce6) for synergistic ferroptosis–photodynamic anticancer treatment, which is approved by the FDA for use in poly(lactic-co-glycolic acid) (PLGA). In the acidic TME, the Fe_3_O_4_-PLGA-Ce6 nanosystem may disintegrate, releasing Ce6 and ferrous/ferric ions. Subsequently, hydroxyl radicals (OH) may be produced and tumor cell ferroptosis can be induced by the Fenton reaction between the liberated ferrous/ferric ions and intracellular excess hydrogen peroxide [223].

### 9.5. Fe NPs in Magnetic Resonance Imaging (MRI)

The introduction of Fe NPs has completely transformed the MRI process, by enabling these NPs to be highly effective as contrast agents [217,224]. Nanosystems are thus able to remarkably tailor the relaxation times of distant water protons, which in turn facilitates the delineation of healthy and affected tissues with more precision [225]. The acidic microenvironment may break down the nanosystem to facilitate Fe_3_O_4_-PLGA-Ce6 liberation of Ce6 and iron ions. Furthermore, in the Fenton reaction, hydroxyl radicals are produced by the interplay between ferrous and ferric ions and the intracellular abundance of hydrogen peroxide, which causes tumor cell ferroptosis [226]. Antibodies that specifically target cancer cells can functionalize Fe NPs, enabling them to interact with malignant cells [227]. This directive method gives the tumor a higher sugar content, so it can be seen in MRI scans which aid in the detection and diagnosis of cancer at an early stage [228]. In the course of atherosclerosis, Fe NPs tend to accumulate in plaques as a result of their hydrophobic qualities. This overload therefore enables the imager to visualize plaque burden as well as assess the risk of cardiovascular disease [229]. Fe NPs, which have surface modifications that allow them to be captured by the filtering functions of the liver and spleen, are also used in imaging [230].

In magnetic resonance imaging (MRI), Fe NPs reduce the transverse relaxation time (T2) of water protons at tissue locations, resulting in the negative contrast performed by Dash et al. (2021). The synthesis of three batches of colloidal Fe NPs with inorganic core diameters of 15.2, 12.0, and 8.8 nm are relatively monodisperse (size dispersion <10%). The 15.2 nm Fe NPs, when dissolved in chloroform and deionized water, exhibit excellent durability against oxidation for more than five months [231]. Liu et al. (2014) synthesized Fe NPs using glutathione (GSH). GSH, an anti-oxidant that is widely present in the human body, and tetrakis(hydroxymethyl)phosphonium chloride (THPC), a reducing agent, are used in this straightforward, one-step reduction. The obtained 3.72 ± 0.12 nm-diameter GSH-IO NPs show a low r^2^ relaxivity (8.28 mm^−1^s^−1^) and r^2^/r^1^ ratio (2.28), both of which are crucial for T1 contrast agents, due to their excellent water-dispersibility and low magnetism [232].

### 9.6. Applications of Fe NPs in Hyperthermia

The hyperthermia process involves increasing the temperature of the selected tissues of the body to induce cellular damage or death, and serves as a means of application in cancer treatment [233]. The Fe materials have been given a lot of attention in this field because of their unique magnetic properties, allowing heating when exposed to a varying magnetic field [234]. Fe nanocomposites have been designed by adding Fe NPs into different matrices, e.g., polymers, and carbon-based materials. Such nanocomposites feature much-improved magnetic properties and stability, leading to accurate heating control during hyperthermia [235]. New studies are focusing on green synthesis methods and on the optimization of the nanocomposite properties to enhance their performance and biocompatibility [236]. Tian et al. (2018) designed magnetic mesoporous silica NPs (MMSN@P(NIPAM-co-MAA)), a thermo-responsive copolymer-coated therapeutic platform for magnetic hyperthermia and chemotherapy. The DOX-loaded NPs had a synergistic effect that enhanced the disintegration of cancer cells by combining magnetic hyperthermia treatment with chemotherapy. As a result, MMSN@P(NIPAM-co-MAA) NPs provide a lot of promise for treating cancer [237]. Yang et al. (2015) fabricated WS_2_ nanosheets that were covered with a mesoporous silica shell that contained polyethylene glycol (PEG) after their surface had been pre-adsorbed with Fe NPs by self-assembly. The development of multifunctional nanoscale theranostics based on two-dimensional transition metal dichalcogenides (TMDCs), like WS_2_, has great potential for the treatment of cancer via multimodal imaging-guided combination therapy [238].

### 9.7. Applications of Fe NPs in the Food Industry

The use of iron (Fe)-containing substances has been attracting interest for their possible role as active polymer constituents in the food packaging industry. Active packaging implies packaging systems developed with the primary goal of extending food products’ shelf life and of providing good quality by interacting with the internal conditions of the package [239,240]. The iron-containing structures can exhibit multiple functions in active packaging systems such as preservation, the use of antimicrobial properties, or by additionally enriching the content of the packaged food [241]. Oxygen scavengers based on iron can be added to packaging materials to attain an efficient removal of oxygen from the headspace. This will therefore prevent or delay oxidative reactions like lipid oxidation, which can contribute to food rancidity and a drop in its quality [242,243]. In general, iron-based scavengers involve the use of iron powder that reacts with oxygen to give rise to iron oxides. Thus, organic scavengers can be put into the films or sachets of the packaging [243]. The particles made of Fe NPs or ions exhibit antibacterial properties against a wide range of foodborne pathogens and spoilage microorganisms. Such Fe-bearing compounds can interfere with the microbial membranes and enzymatic activity, induce oxidative stress, and therefore lead to the destruction of the microorganisms [244]. These can be added to the packaging items of food products to enhance their shelf life by minimizing microbial growth [245]. Fe is a micronutrient that is crucial in different physiological functions including oxygen transport and metabolism in the human body. Nevertheless, iron deficiency is a dominant nutritional problem among people on a global level [246]. The packaging materials that have Fe NPs or ions will release iron into the food sealed inside the package, thereby fortifying the food product with this important nutrient. Such methods can be very useful in foods meant for iron deficiency-ridden populations, for example, babies and pregnant women [247]. Additionally, packaging materials can also be dosed with iron ions to be used as coloring agents to check for the freshness of packaged food products [248].

## 10. Future Perspectives and Challenges

Fe NPs have a bright future ahead of them, after demonstrating a promising future in several industries. Despite their potential, the utilization of Fe NPs is fraught with difficulty. The potential toxicity of these compounds for living organisms and ecosystems is of major concern. Extensive research efforts are being made to develop the appropriate safety precautions and grasp the results of the effect of Fe NPs on the environment and human health [249]. Another difficulty of Fe NPs is their reactivity and stability. Their functionality may be affected and use may become restricted due to their tendency to oxidize and their aggregation ability. Researchers are trying to develop methods to improve the stability and monitoring responsiveness of Fe NPs. There is a need to design the cost-effectiveness and expansion of production of Fe NPs on a commercial scale. It is necessary to develop effective and viable techniques that satisfy the increasing demand and thus enable the application of these NPs to a wide range of applications [250]. The key challenges in the utilization of Fe NPs for antifungal applications are shown in Figure 21.

## 11. Emerging Trends and Future Directions in Optimizing Fe NPs for Antifungal Applications

As summarized above, Fe NPs have demonstrated considerable potential across a variety of sectors of life, including electronics, environmental science, and the medical area. Recently, there has been a surge in research on the synthesis and applications of Fe NPs, particularly in the field of antifungal treatments. Therefore, it is important to understand the challenges and opportunities that lie ahead in order to improve these NPs for safe and effective therapeutic purposes [251]. The emerging trends of the Fe NPS are shown in Figure 22.

The development of innovative synthesis techniques is one of the main future directions for the use of Fe NPs in antifungal applications. Innovative methods for producing Fe NPs with regulated surface changes, size, and shape have been investigated. Researchers can improve the stability, general therapeutic efficacy, and dispersibility of these NPs by adjusting these parameters. Moreover, research is being conducted to explore the application of economical and ecologically friendly synthesis methods to enhance scalability and lower the total production expenses [252].

However, Fe NPs can be functionalized to improve their antifungal characteristics. Applying different coatings, such as biomolecules or polymers, to the surface of these NPs may enhance their stability, drug-loading capability, and targeting ability. Antifungal compounds can also be released under controlled conditions owing to functionalization, which maximizes their effectiveness and reduces any negative side effects. To further improve their specificity, responsive components can be added to the nanoparticle structure to allow triggered drug release in response to fungal indicators [253].

Even with bright prospects, a few issues must be resolved. However, the toxicity of Fe NPs is a major obstacle. Comprehensive toxicity studies must be conducted to evaluate the possible adverse effects and safety. To guarantee the long-term efficacy of these NPs, it is necessary to assess their stability in biological systems [254]. Optimization of the dosage and administration route is another challenge. The optimal concentration and frequency of Fe NP delivery must be established to obtain the best antifungal results while reducing side effects. Furthermore, investigating different delivery methods, such as topical or inhalation delivery, can yield more effective and specialized treatment choices for fungal infections [255]. Moreover, it is crucial to develop uniform testing procedures and recommendations for evaluating the antifungal properties of Fe NPs. Maintaining uniformity in the testing procedures might make it easier to compare the findings from various studies and improve the accuracy of the collected data. Future directions for the optimization of Fe NPs for antifungal mechanisms are shown in Figure 23.

### 11.1. Overcoming Challenges and Barriers regarding the Translation of Fe NPs into Clinical Settings

A plethora of research has been documented regarding Fe NPs and their potential utilization in medicine. Fe NPs have great potential, and to guarantee the effective integration of Fe NPs into therapeutic applications several issues must be resolved before the integration of Fe NPs into therapy [256]. The primary obstacle faced during the production of Fe NPs is the controlled or tunable dimensions and shape, as these characteristics have a substantial influence on their functions and effectiveness. For successful and effective translation of Fe NPs, it is vital to achieve reliable, consistent, robust, and scalable synthetic methods for yielding fine-quality Fe NPs. Moreover, the stability and biocompatibility of Fe NPs should be investigated comprehensively because it is critical to provide safe materials for clinical applications. Additionally, comprehensive toxicological studies need to be carried out to validate the biocompatibility of Fe NPs and to record any potential hazards [257]. Another obstacle in using Fe NPs is understanding their long-term fate and biodegradability in the human body. Fe NPs may degrade and release Fe ions that can interact with and affect the function of cellular components [258]. Furthermore, the successful application of Fe NPs in clinical settings depends on the development of efficient delivery techniques. To guarantee accurate delivery to the target site and decrease off-target impacts, selectivity and specificity must be optimized to achieve a maximum concentration of Fe NPs at the target site [259]. To avoid side effects and to ensure the safe and efficient utilization of Fe NPs to acquire better outcomes during patients’ therapy, interdisciplinary collaboration between scientists, engineers, physicians, and regulatory agencies is necessary to overcome these obstacles [260].

### 11.2. Regulatory and Commercialization Prospects for Fe NP-Based Antifungal Products

Fe NPs have been shown to be remarkable and promising substitutes in many disciplines, including medicine. The use of Fe NPs in antifungal applications can offer both opportunities and challenges for the future and should be studied from a commercialization and regulatory standpoint. A comprehensive evaluation of the safety and effectiveness of Fe NP-based antifungal treatments is becoming necessary before they can be used, from a regulatory perspective. Regulatory bodies are essential for safeguarding public health because they guarantee the efficacy and safety of new technologies. Therefore, to fully comprehend the possible risks associated with the use of Fe NPs, both for humans and the ecosystem, substantial research and testing must be conducted [261]. Nevertheless, despite regulatory obstacles, there are encouraging commercialization possibilities for iron nanoparticle-based antifungal medicines. They can be used in a variety of applications including wood preservation, food production, medicinal procedures, and agricultural products [262,263]. Fe NPs-based antifungal medicines have the potential to treat drug-resistant fungal infections in the medical field. Their applications range from lotions, ointments, and coatings to various formulations for the treatment of both systemic and superficial fungal illnesses. However, several obstacles must be overcome for effective commercialization. These include increasing the production volume, developing affordable manufacturing procedures, guaranteeing the stability and homogeneity of Fe NPs, and refining the synthesis techniques. Concerns regarding compatibility with other materials used in the corresponding applications, possible toxicity, and long-term stability must also be considered by product creators [264].

Finally, given their potential applications in medicine, the food industry, and agriculture, the prospects for Fe NPs-based antifungal solutions are promising. However, the first stages of successful implementation involve regulatory considerations and address safety, environmental effects, and commercialization issues. Through thorough assessment, investigation, and cooperation between regulatory agencies, businesses, and academia, Fe NPs may prove to be useful tools in the fight against fungal diseases while posing the fewest hazards to the environment and public health [265]. The regulatory considerations and commercialization prospects for Fe NP-based antifungal properties are illustrated in Figure 24.

## 12. Implications and Potential Impact of Optimized Fe NPs in Biomedical Applications

Fe NP optimization shows great promise for a variety of medicinal applications, owing to their antifungal characteristics. Therefore, the incorporation of Fe NPs has the potential to transform medicine by offering novel and practical methods for the therapy of fungal diseases and improving patient outcomes [266]. The development of optimized Fe NPs has several consequences for biomedical applications of Fe NPs. First, treating fungal infections, which present a severe threat in clinical settings, can be made easier by using these NPs to improve their antifungal potential. During therapy, it is required to enhance on-target action and elimination of fungal pathogens, and these NPs can minimize the necessity for traditional antifungal medications that have limited effectiveness or which offer drug resistance [267]. In addition to the antifungal characteristics, Fe NPs have other advantages when used in biological applications. For example, Fe NPs can be applied as medication delivery vehicles, allowing therapeutic molecules to be released precisely at the infection site [268]. Furthermore, the optimization of Fe NPs for biomedical application may enable their integration into a range of implants and medical equipment, including catheters, prosthetic devices, and wound dressings. Fe NPs can be applied to render the surfaces of such medical devices or equipment unsuitable for or unfriendly to fungal colonization, thus lowering the risk of infection and improving patient safety [185]. In general, the biomedical field can undergo a transformation if Fe NPs are optimized before their incorporation into biomedical applications. Finally, by utilizing Fe NPs’ antifungal characteristics and investigating their multifunctionality, future efforts can open new avenues for the therapy of fungal infections, improve onsite drug delivery, and enhance the safety and efficacy of different medical procedures. Further investigations and innovations in this field are essential to properly understand and optimize the possible adverse effects of optimized Fe NPs [269].

## 13. Closing Remarks and Suggestions for Future Research

In summary, the optimization of Fe NPs has considerable potential for biomedical applications in terms of revealing their antifungal capabilities. These characteristics reduce the possibility of harmful effects on healthy cells, allow for focused drug administration, and promote synergistic action with other antifungal drugs [270]. Several investigations have shown that optimized Fe NPs can be used to successfully block fungal growth and biofilm production both in vitro and in vivo. These results demonstrated the great potential of Fe NPs as a possible adjuvant or substitute treatment for fungal infections. Moreover, the application of Fe NPs can help reduce the issue of antifungal medication resistance, which is becoming increasingly pressing. Future studies in this area should focus on resolving specific issues to improve the area of research. This could involve determining the precise mechanisms underlying the antifungal action of Fe NPs, improving their stability and reproducibility through manufacturing, and investigating their suitability for various drug delivery systems. Furthermore, in vivo investigations should be conducted to assess the pharmacokinetics, safety, and long-term effectiveness of treatments using Fe NPs [271]. Figure 25 illustrates the future direction and prospective research on the optimization of Fe NPs.

## 14. Conclusions

In summary, thorough research and the optimization of Fe NPs for antifungal characteristics hold enormous potential for the development of medicinal applications. An in-depth investigation emphasized various strategies used to improve the antifungal effectiveness of Fe NPs, such as size management, surface modification, and synergistic approaches with other therapeutic agents. These enhancements not only improve the antifungal efficacy of Fe NPs but also ensure their biocompatibility and decreased cytotoxicity, opening the door for their safe and effective use in a variety of biomedical applications. The evolving research environment surrounding Fe NPs emphasizes their potential to solve critical difficulties faced by fungal infections in healthcare. Fe NPs provide a diverse platform for addressing fungal infections while minimizing the detrimental effects on host cells, ranging from targeted medication delivery to new wound dressings and implant coatings. Furthermore, the ecologically benign techniques for nanoparticle manufacturing addressed herein underline the need for sustainable synthetic processes. However, as we learn more about the antifungal properties of Fe NPs, it becomes clear that further research is needed to optimize their efficacy. In-depth investigations of the mechanisms of action, long-term safety evaluations, and the development of standardized techniques for Fe NP synthesis and characterization should be part of future research priorities. Collaboration among researchers from several disciplines, such as materials science, microbiology, and medicine, is critical for realizing the full potential of Fe NPs for medicinal applications. In summary, the path to harnessing the antifungal characteristics of Fe NPs is still in the early stages of development. With ongoing research efforts, we are poised to see remarkable advancements in healthcare, where optimized Fe NPs have emerged as valuable tools in the fight against fungal infections, ultimately improving patient care and pushing the boundaries of biomedical innovation. Fe NPs are fascinating nanomaterials with several applications in material research, environmental remediation, and medicine. To optimize the potential of Fe NPs, one must first understand their formation and function.

## Figures and Tables

**Figure 1 pharmaceutics-16-00645-f001:**
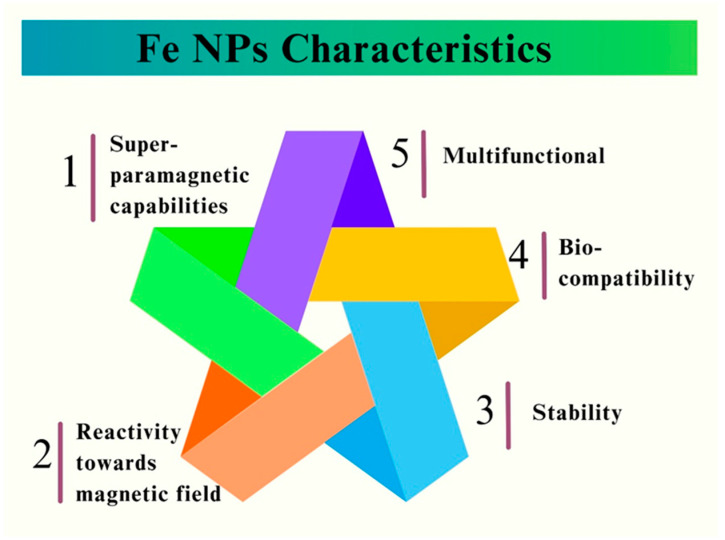
Characteristic features of Fe NPs.

**Figure 2 pharmaceutics-16-00645-f002:**
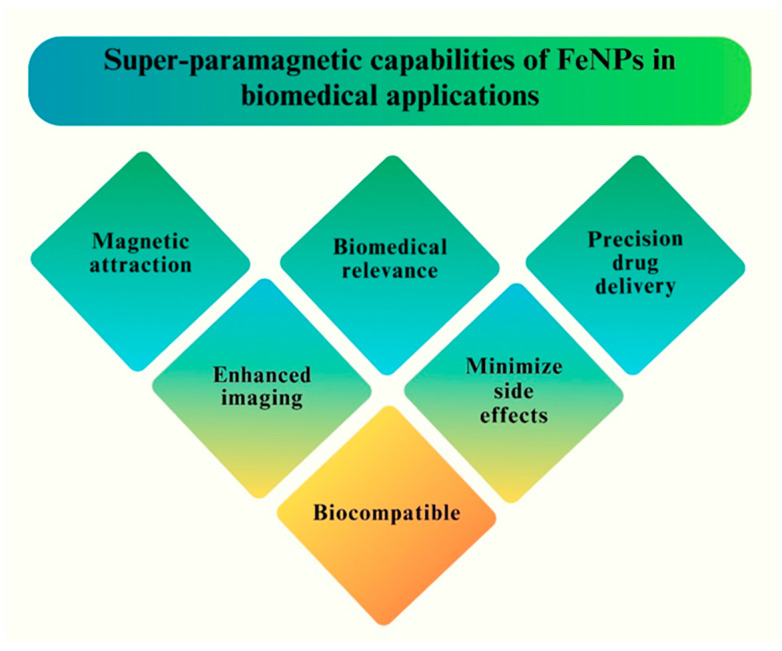
Super-paramagnetic capabilities of Fe NPs in biomedical applications.

**Figure 3 pharmaceutics-16-00645-f003:**
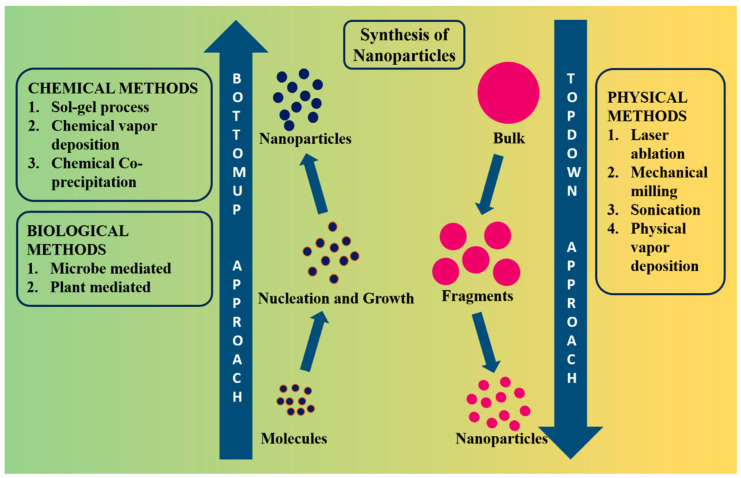
Schematic representation of most common approaches for the synthesis of NPs.

**Figure 4 pharmaceutics-16-00645-f004:**
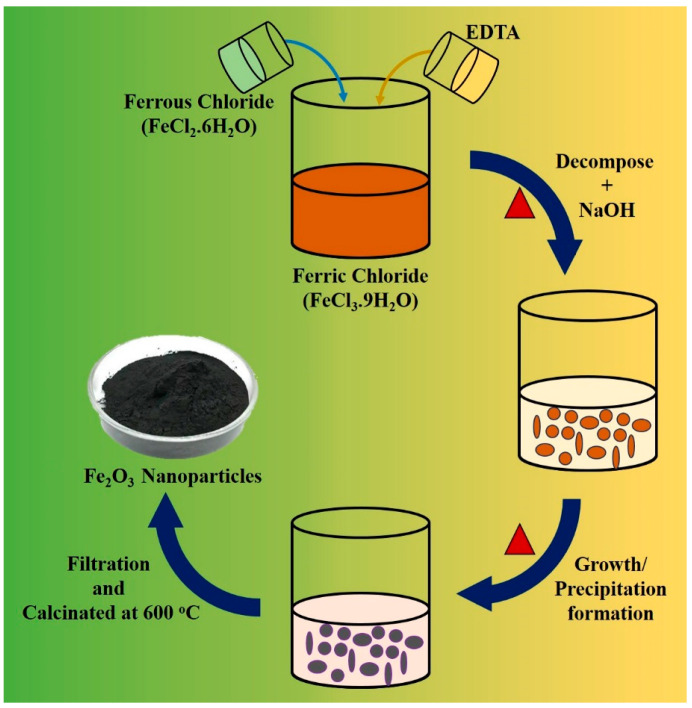
Schematic representation for the synthesis of Fe NPs through the co-precipitation method.

**Figure 5 pharmaceutics-16-00645-f005:**
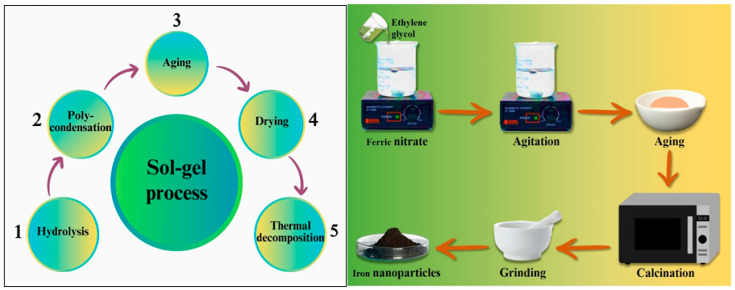
Schematic representation of the sol–gel process for the synthesis of Fe NPs.

**Figure 6 pharmaceutics-16-00645-f006:**
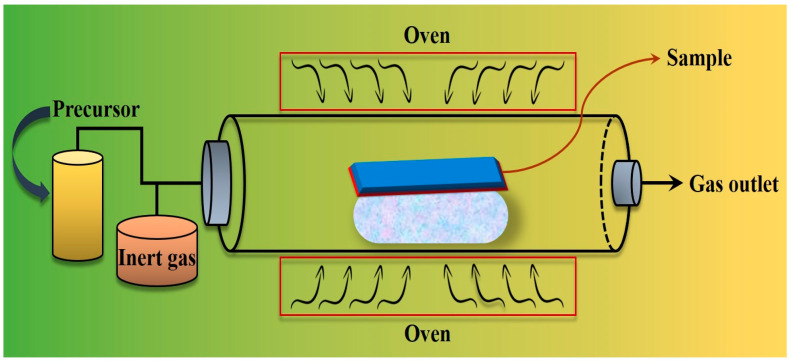
An overview of the synthesis of Fe NPs through CVD.

**Figure 7 pharmaceutics-16-00645-f007:**
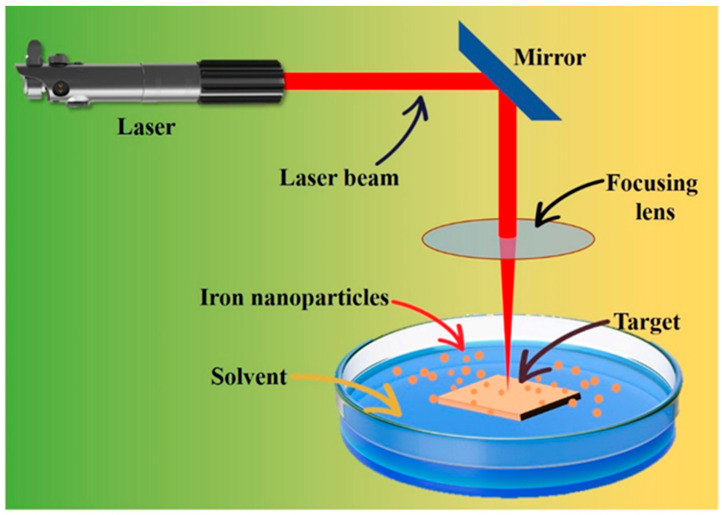
Synthesis of Fe NPs by process of laser ablation.

**Figure 8 pharmaceutics-16-00645-f008:**
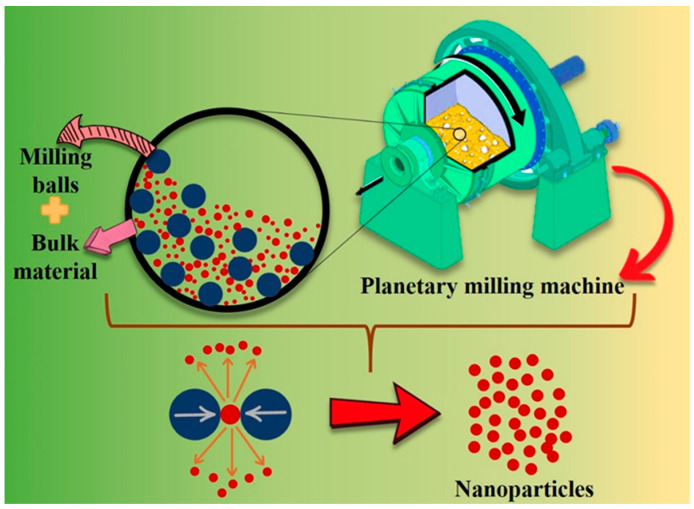
Synthesis of Fe NPs through mechanical milling.

**Figure 9 pharmaceutics-16-00645-f009:**
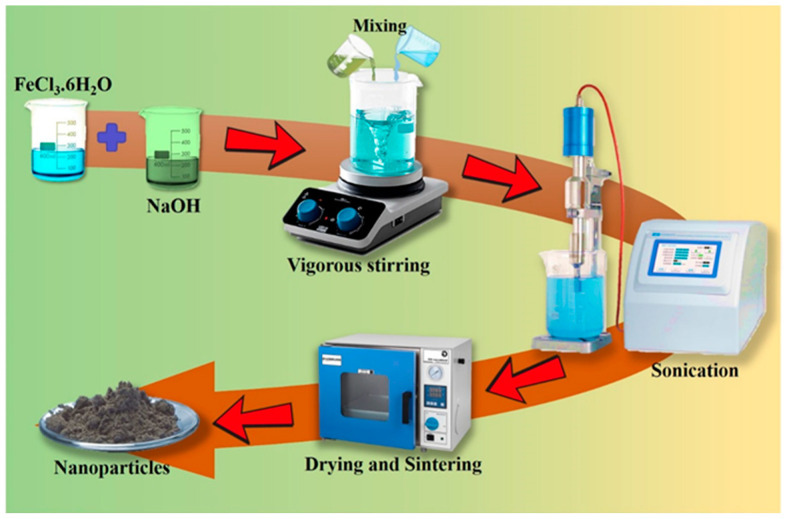
Illustration of sonication methodology routinely employed for the synthesis of Fe NPs.

**Figure 10 pharmaceutics-16-00645-f010:**
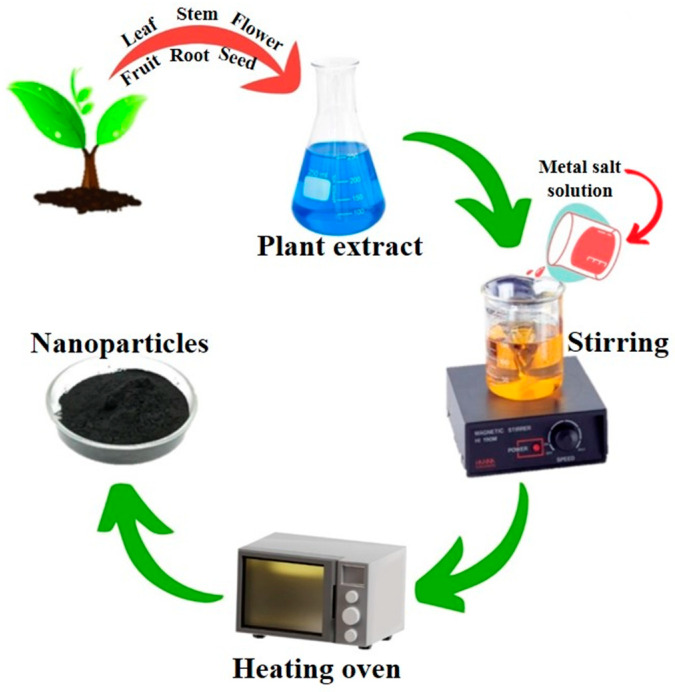
Stepwise schematic illustration for plant-mediated synthesis of Fe NPs.

**Figure 11 pharmaceutics-16-00645-f011:**
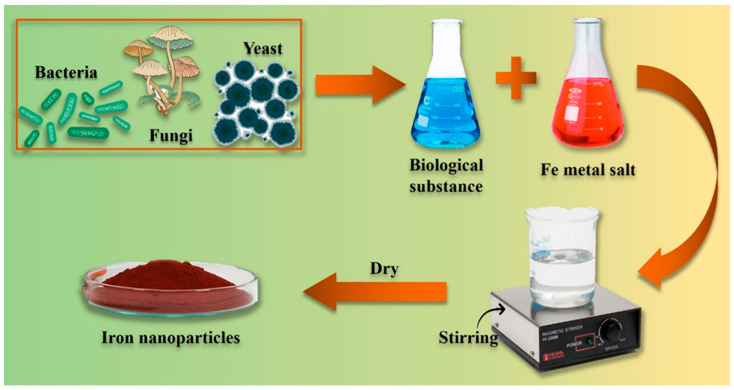
A typical overview of the microbial-mediated synthesis of Fe NPs.

**Figure 12 pharmaceutics-16-00645-f012:**
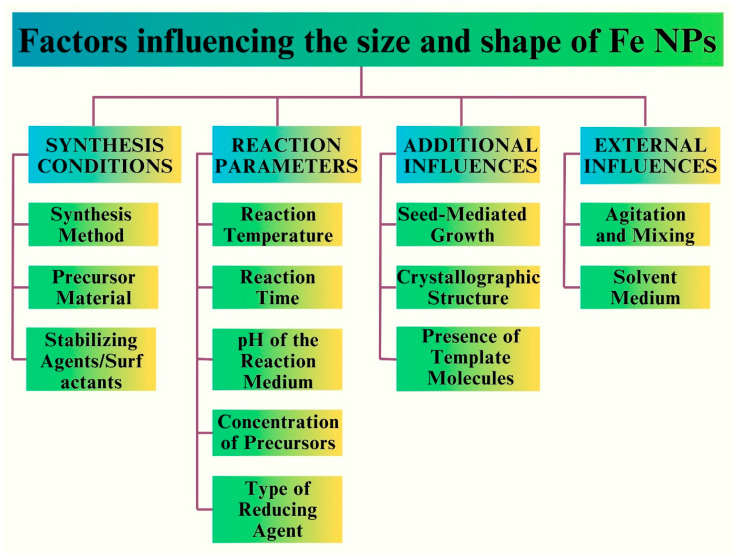
Key factors influencing the size and shape of Fe NPs.

**Figure 13 pharmaceutics-16-00645-f013:**
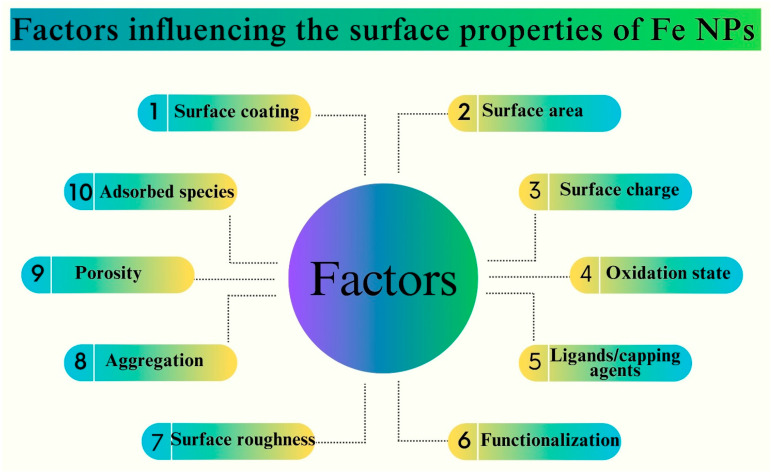
A summary of key nuts and bolts influencing the surface properties of Fe NPs.

**Figure 14 pharmaceutics-16-00645-f014:**
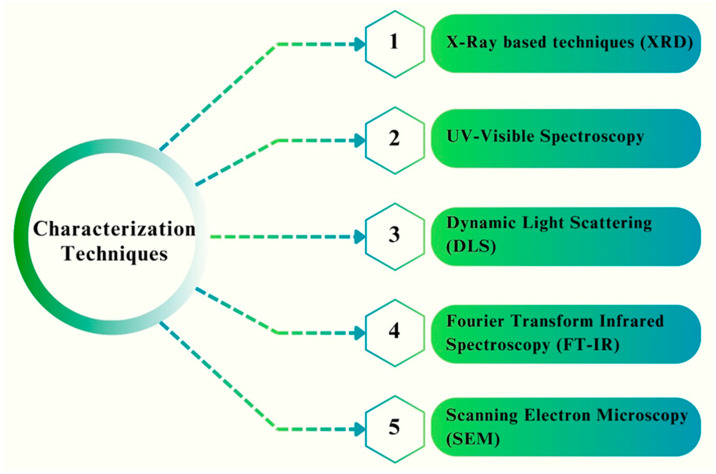
Routinely employed characterization techniques for analyzing the physiochemical properties of Fe NPs.

**Figure 15 pharmaceutics-16-00645-f015:**
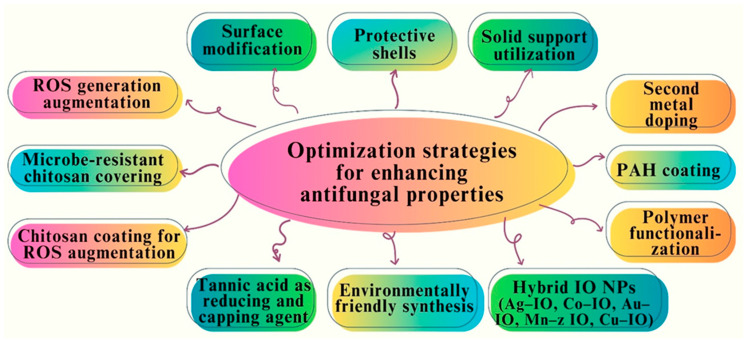
Fundamental strategies for the optimization and enhancing of antifungal properties of Fe NPs.

**Figure 16 pharmaceutics-16-00645-f016:**
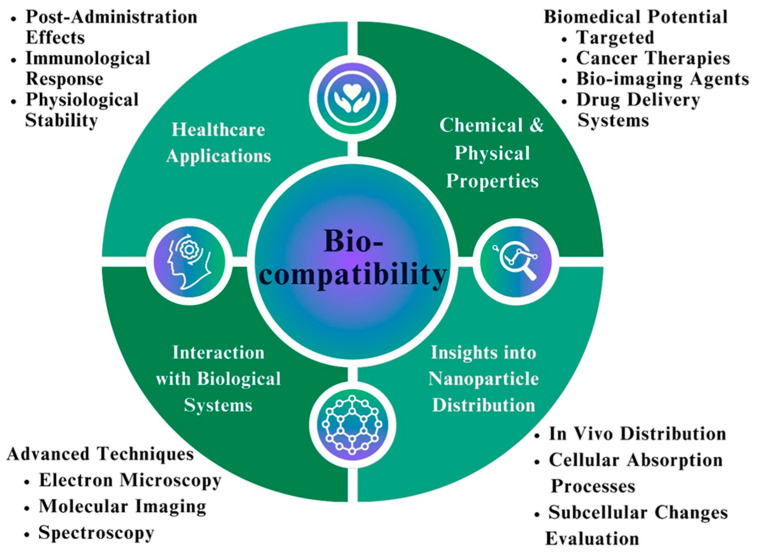
A schematic representation of biocompatibility of Fe NPs.

**Figure 17 pharmaceutics-16-00645-f017:**
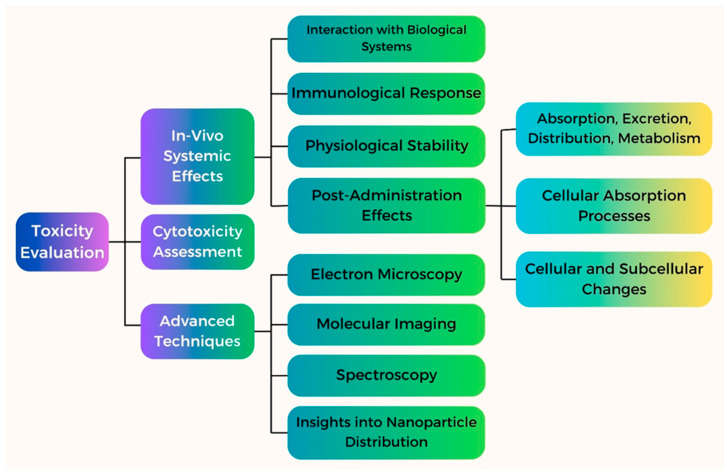
Representing the toxicity evaluation of Fe NPs.

**Figure 18 pharmaceutics-16-00645-f018:**
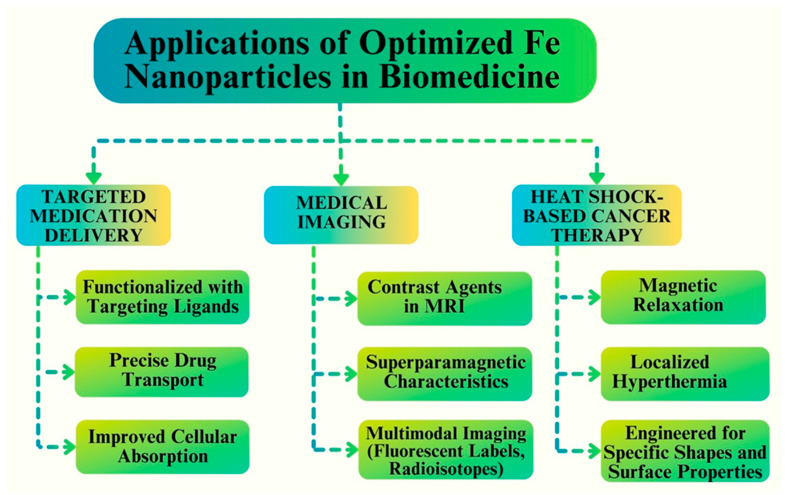
Applications of optimized Fe NPs in biomedicine.

**Figure 19 pharmaceutics-16-00645-f019:**
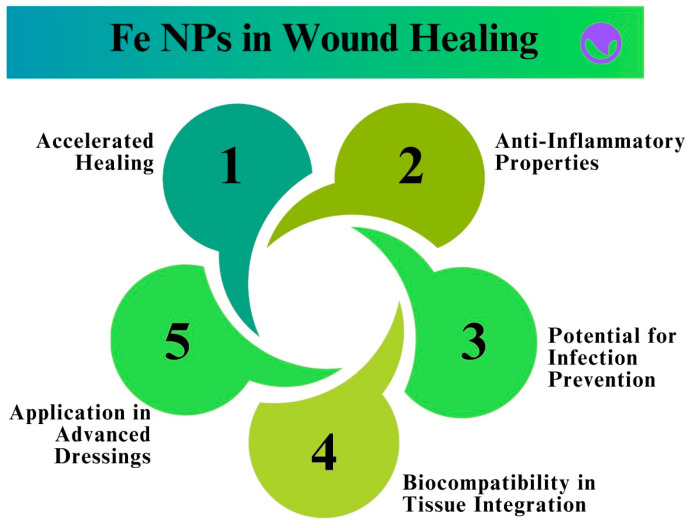
Fundamental aspects of the biocompatibility of Fe NPs in the wound healing process.

**Figure 20 pharmaceutics-16-00645-f020:**
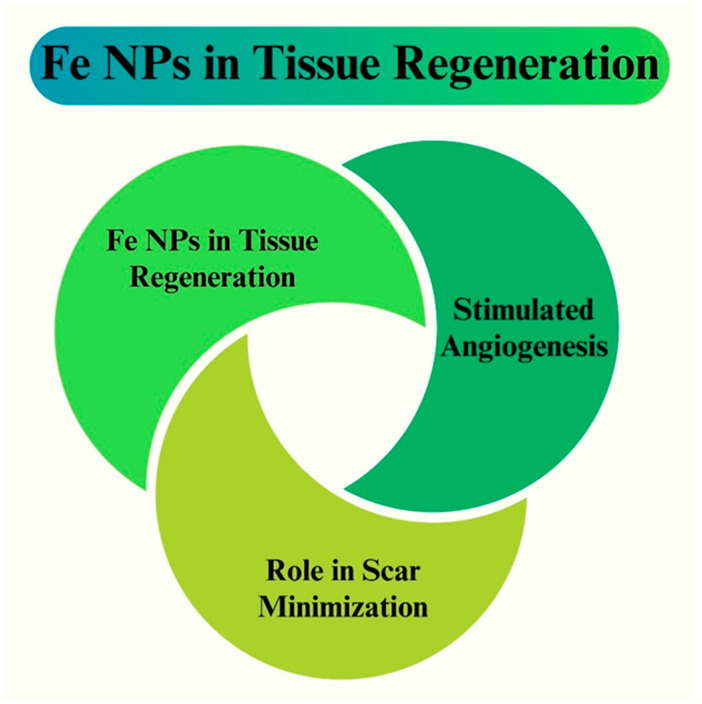
Biocompatibility of Fe NPs in the tissue regeneration process.

**Figure 21 pharmaceutics-16-00645-f021:**
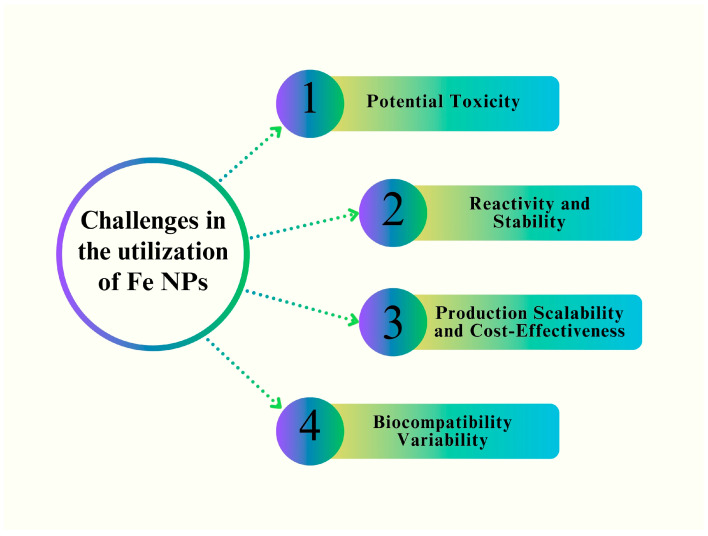
Key challenges in the utilization of Fe NPs for antifungal applications.

**Figure 22 pharmaceutics-16-00645-f022:**
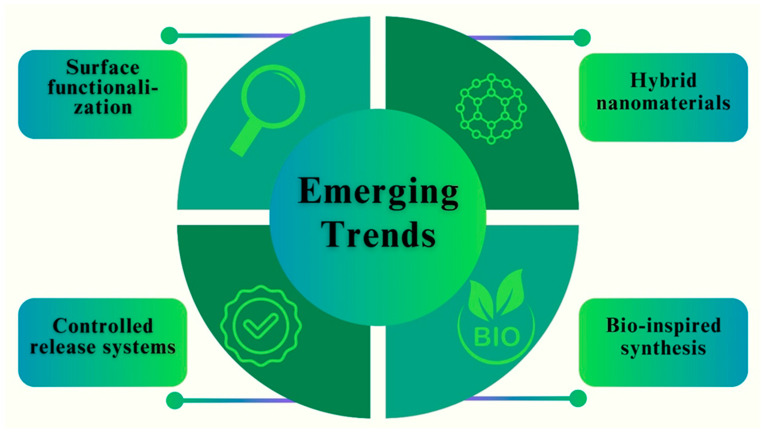
Emerging trends in optimizing Fe NPs for antifungal applications.

**Figure 23 pharmaceutics-16-00645-f023:**
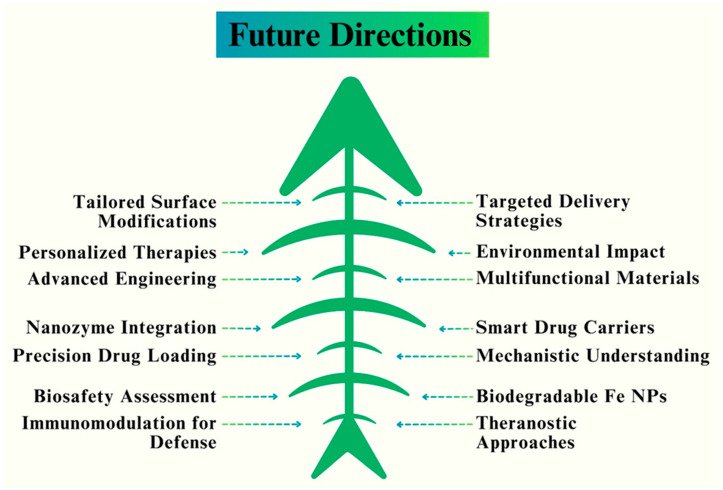
Future directions in optimizing Fe NPs for antifungal applications.

**Figure 24 pharmaceutics-16-00645-f024:**
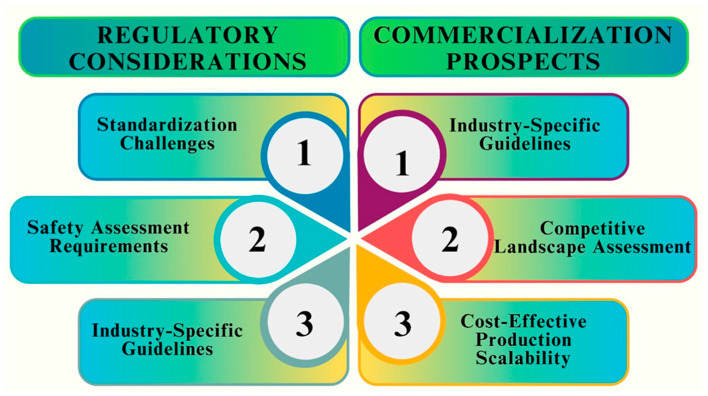
Regulatory considerations and commercialization prospects for Fe NP-based antifungal products.

**Figure 25 pharmaceutics-16-00645-f025:**
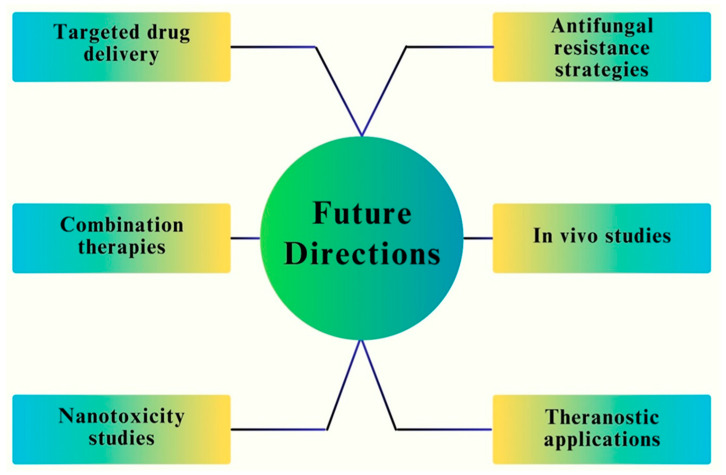
Recommendation for future research into the optimization of Fe NPs.

**Table 1 pharmaceutics-16-00645-t001:** A summary of key synthetic strategies, biological evaluations, and structural properties of Fe NPs.

Nanoparticle	Synthesis Method	NP Size (nm)	Surface Functionalization	Antifungal Activity/Fungal Strains	Biocompatibility Assessment	Ref
Fe_2_O_3_	Chemical co-precipitation	100–130	Oleic acid functionalization, tagged with Itraconazole and encapsulation with PHB.	Against *C. albicans* at the highest concentration of 20 µg/mL.	Encapsulation with Polyhydroxybutyrate (PHB).	[74]
Fe_2_O_3_	Green Method	10–30	Iron oxide NPs (1%) were fabricated using tannic acid in an alkaline medium.	*Trichothecium roseum*, *Cladosporium herbarum*, *Penicillium chrysogenum*	The highest inhibition was caused by *P. chrysogenum* (28.67 mm) followed by *A. niger* (26.33 mm).	[75]
Chitosan-coated Fe_2_O_3_	Co-precipitation	10.4 ± 4.9	Antimicrobial activity was tested by agar well diffusion and analyzed by measuring the diameter of the inhibition zone.	Against *Aspergillus niger (A. niger)* and *Fusarium solani (F. solani.*	Inhibition zone of chitosan-coated Fe_2_O_3_ NPs = 14.5 to 18.5 mm.	[76]
Fe_2_O_3_	Biosynthesis	30.98	-	*A. brasiliensis*, *A. alternata*, *F. oxysporum*, *C. albicans*	Diameters of inhibition zone, *A. brasiliensis* = 36.08 ± 1.37, *A. alternata* = 27.59 ± 1.32, *F. oxysporum* = 26.11 ± 1.11, *C. albicans* = 53.67 ± 3.18	[77]
Fe_2_O_3_	Green Synthesis	20–86	10 g of *Wedelia urticifolia* leaves was utilized for surface functionalization.	Against *Candida albicans.*	-	[78]
Ag, Cu, Fe, Zn	Biosynthesis	18.33	Ginger and garlic extract.	*C. albicans*	Inhibition one diameter = 7 mm	[79]
Fe	Green Synthesis	50	Aqueous leaves extract of *Plumeria obtuse.*	*A. niger* and *S. commune.*	Doxorubicin and Cisplatin were used as standards while 2.1 ± 0.01% hemolysis is shown by Fe NPs.	[80]
Fe_2_O_3_	Green synthesis	76	Microalgal proteins, carbohydrates, and polyphenols are responsible for bio fabrication.	*Fusarium oxysporum*, *Fusarium tricinctum, Fusarium maniliforme, Rhizoctonia solani*, and *Phythium* sp.	Inhibition one diameter = 10–25 mm	[81]
Fe_2_O_3_	Green synthesis	38	Agar well diffusion method with week-old fungal cultures grown on potato dextrose medium.	*Aspergillus niger* and *Mucor piriformis*	Activity against *Aspergillus niger* = 16 mm and *Mucor piriformis* = 26 mm	[82]
Fe_2_O_3_/Ag@Fe_2_O_3_	Green synthesis	4 ± 1	Agar well diffusion and macro dilution broth method.	*Candida albicans*	Inhibition zone Fe_2_O_3_ = 10 mm	[83]
SrFeO_3-δ_	Sol–gel method	91.28	Nystatin was used as a standard.	*Candida albicans* and *Aspergillus brasiliensis*	Inhibition zone diameter *Candida albicans* = 7.81 ± 0.91. *Aspergillus brasiliensis* = 20.2 ± 0.05	[84]
Ag–Fe bimetallic	Green synthesis	3–30	Micro dilution.	*C. albican*	MIC value = 62.5 ppm	[85]
Fe NPs	Green Synthesis	40.4	-	*C. albican*	Inhibition zone = 21 mm	[86]
Fe/Cu/Ag	Green synthesis	-	Aflatoxins (B_1_, B_2_, G_1_ and G_2_) standards used.	*Aspergillus flavus* and *A. parasiticus*	-	[87]

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
