# Peer review of "Advances in the Optimization of Fe Nanoparticles: Unlocking Antifungal Properties for Biomedical Applications"

_pharmaceutics, 2024, doi:10.3390/pharmaceutics16050645_

Round 1
Reviewer 1 Report
Comments and Suggestions for Authors
Before publication, the manuscript should be revised.
1. More clinical application of Fe materials should be introduced!
2. The application of Fe materials in imaging and hyperthermia should be addressed and more lasted examples should be introduced!
3. Some related references should be cited, like Chinese Chemical letters, 2023, 34: 107650.
Author Response
- A detailed response to Reviewer No.1's feedback has been attached for further evaluation.
- I have highlighted the changes made for reviewer 1 within the manuscript.

Reviewer 2 Report
Comments and Suggestions for Authors
The manuscript "Advances in the Optimization of Fe Nanoparticles: Unlocking Antifungal Properties for Biomedical Applications", written by Sandhu et al., presents a potential perspective review of iron nanoparticles applied for their antifungal properties in biomedicinal applications. The text is well written, the chapters are easily readable, and adequate recent references accompany the text. The theme of the review is worth pursuing and can have an impact not only on biomedicine but also on other related scientific fields.
I have several comments. Firstly, the volume of the review chapters is not well-balanced. The title and the abstract imply that the review will cover the antifungal properties of Fe NPs and their application in related fields. However, a considerable part of the text covers the synthesis and functionalization of NPs, which was covered many times before in other reviews. Major changes in the structure of the review have to be done according to my opinion. I thus suggest considerably shortening chapters 2 - 5 (now almost half of the overall review) and focusing on chapters 6 and 7. Chapter 6 is especially significant; it is extremely short and does not go into any details. Chapter 7 should also be extended and should cover recent trends and findings. Chapter 8 should focus more on antifungal properties and not on general toxicity. This was also covered many times elsewhere. Chapter 9 should focus more on applying NPs for their antifungal properties. The chapter is now just general.
Author Response
- A detailed response to Reviewer No.2's feedback has been attached for further evaluation.
- I have highlighted the changes made for reviewer 2 within the manuscript.

Reviewer 3 Report
Comments and Suggestions for Authors
Fe-based nanoparticles are widely used in different areas of human activity, including magneto-optical device elaboration, biomedicine, drug delivery platform design, eco-friendly remediation, and more.
Along with thorough surveying Fe-NP preparation techniques, the authors put their focus on antifungal endeavor of the nanoparticles as the own items and the drug vehicles that is extremely when creating both innovative biomaterials and active food packaging.
The topics of this review, submitted obviously fall within the scope of the Pharmaceutics MDPI. With suitable terminology and reasonable argumentation, the manuscript shows clearly the resisters of Fe (Fe2O3) NPs against fungi activity. The manuscript abstract reflects the general issues of the paper adequately. The bibliography is quite relevant to this study and the illustrations (the tables and the figures) are executed in the clear and accurate manner with the quite coherent interpretation.
Among the shortcomings observed in the text, it is worth noting:
• Sometimes, the authors do not separate the terms "Fe NP-based" which is true for iron nanoparticles with a zero valency and iron oxide nanoparticles with mixed cation valency, namely [Fe(II) and Fe(III)], see, for example, the caption of Fig. 4.
• Authors are invited to emphasis that besides the very important area of Fe-NP applications in biomedicine and pharmaceutical industry, the Fe-containing entities can be successfully implemented as the active packaging constituents in food industry as well.
Finalizing, it is appropriate to recommend this paper for the next Edition execution after making the above minor corrections.
Author Response
- A detailed response to Reviewer No.3's feedback has been attached for further evaluation.
- I have highlighted the changes made for reviewer 3 within the manuscript.

Reviewer 4 Report
Comments and Suggestions for Authors
Figure 3 Should be "bottom up approach" instead of "bottomup approach".The publication is very extensive. It is almost a chapter in a book while the conclusions are minimal. This ratio is not classical. Maybe it would have been better to focus on some single aspect.Nevertheless, the work is correct in terms of content as well as technically. No major flaws are found.
Comments on the Quality of English LanguageOK
Author Response
- A detailed response to Reviewer No.4's feedback has been attached for further evaluation.
- I have highlighted the changes made for reviewer 4 within the manuscript.

Round 2
Reviewer 2 Report
Comments and Suggestions for Authors
The authors answered all my comments and improved the structure and text flow in the revised version of the manuscript. I do not have any additional comments.